# A major role of class III HD-ZIPs in promoting sugar beet cyst nematode parasitism in *Arabidopsis*

**Xunliang Liu, Melissa G. Mitchum** ⓘ *

Department of Plant Pathology and Institute of Plant Breeding, Genetics, and Genomics, University of Georgia, Georgia, United States of America

* melissa.mitchum@uga.edu

## Abstract

Cyst nematodes use a stylet to secrete CLE-like peptide effector mimics into selected root cells of their host plants to hijack endogenous plant CLE signaling pathways for feeding site (syncytium) formation. Here, we identified *ATHB8*, encoding a HD-ZIP III family transcription factor, as a downstream component of the CLE signaling pathway in syncytium formation. *ATHB8* is expressed in the early stages of syncytium initiation, and then transitions to neighboring cells of the syncytium as it expands; an expression pattern coincident with auxin response at the infection site. Conversely, *MIR165a*, which expresses in endodermal cells and moves into the vasculature to suppress HD-ZIP III TFs, is down-regulated near the infection site. Knocking down HD-ZIP III TFs by inducible over-expression of *MIR165a* in *Arabidopsis* dramatically reduced female development of the sugar beet cyst nematode (*Heterodera schachtii*). HD-ZIP III TFs are known to function downstream of auxin to promote cellular quiescence and define stem cell organizer cells in vascular patterning. Taken together, our results suggest that HD-ZIP III TFs function together with a CLE and auxin signaling network to promote syncytium formation, possibly by inducing root cells into a quiescent status and priming them for initial syncytial cell establishment and/or subsequent cellular incorporation.

**Data Availability Statement:** Raw sequence data have been submitted to the National Center for Biotechnology Information Sequence Read Archive under BioProject number PRJNA1054488.

## Author summary

Plant-parasitic cyst nematodes are one of the most damaging pathogens impacting crop production. These nematodes establish a permanent feeding site called a syncytium to withdraw nutrients from plant roots, causing plant malnutrition and reduced crop yield. The syncytium is induced by nematode stylet-secreted effectors, many of which hijack innate plant developmental programs to convert a normal plant cell to a syncytial cell. One such group of effectors functions by mimicking plant CLE peptide hormones, which are important in plant cell fate determination, and play a crucial role in syncytium formation. In this study, we show that nematode secreted CLE-like effectors induce the expression of the *ATHB8* gene, a member of the Class III HD-ZIP transcription factor family, at early stages of syncytium formation. Reducing the level of HD-ZIP III family genes

**Funding:** This work was supported by a grant from the National Science Foundation (grant no. IOS-1456047 to M.G.M. https://new.nsf.gov/funding/opportunities/plant-biotic-interactions) and the University of Georgia Office of the President and Georgia Agricultural Experiment Stations (to M.G.M. https://president.uga.edu/.) The funders did not play any role in the study design, data collection and analysis, decision to publish, or preparation of the manuscript.

**Competing interests:** The authors have declared that no competing interests exist.

compromised the nematode's ability to feed on plants, indicating that HD-ZIP III transcription factors are important for cyst nematode parasitism.

## Introduction

Obligate sedentary endoparasitic cyst nematodes (CNs) require a reliable and sustained local source of nutrition within host roots to complete their life cycle. They achieve this by establishing a unique and highly specialized feeding site called a syncytium in host roots. The syncytium forms when an infective juvenile selects a single root cell, usually a procambium or pericycle cell [1–4], in the vasculature of a host root, and secretes a suite of effectors through a stylet to induce massive re-programming of the selected host cell [5,6]. The syncytium expands along the vascular cylinder by gradually incorporating neighboring cells by partial cell wall dissolution and protoplasmic fusion [2,7–9]. Researchers have strived to understand the signaling events regulating the highly orchestrated process of syncytium formation. These efforts have shed light on the critical role of CN effector-mediated modulation of phytohormone and peptide hormone signaling pathways in syncytium formation, most notably through the modulation of auxin and CLE signaling pathways [10].

At the early stage of infection, perturbations to auxin homeostasis are key to syncytium establishment. A local accumulation of auxin is pivotal for syncytium initiation and is associated with changes in auxin biosynthesis, polar transport, and signaling [11–15]. Several CN effectors have been identified to directly modulate auxin accumulation and signaling at the infection site. The beet cyst nematode (BCN) tyrosinase-like effector increases auxin content and promotes BCN susceptibility when ectopically expressed in *Arabidopsis* [12]; the BCN effector 19C07 directly interacts with *Arabidopsis* Like-AUX1 3 (LAX3) auxin influx protein and is likely to increase LAX3 activity and promote auxin flow into the syncytium and adjacent cells [14]; while the BCN effector 10A07 is translocated into the plant cell nucleus after being phosphorylated by plant protein kinase IPK (AT2G37840) and interacts with Indoleacetic Acid-Induced Protein 16 (IAA16), a suppressor of Auxin Responsive Factors (ARFs) to possibly release corresponding ARFs for auxin signaling [16]. Host genes related to auxin biosynthesis, distribution, and signaling are also modulated by CN at the infection site. *YUCCA4* (*YUC4*), encoding a member of YUCCA family enzymes that mediates a speed limiting step in the main auxin biosynthesis pathway, is up-regulated at the CN infection site, along with a few other auxin biosynthesis enzymes [13,17]. Auxin efflux *PIN1* and *PIN7* genes are down-regulated at the early feeding site to prevent out flow of auxin from the initial syncytial cell, while PIN3 and PIN4 proteins are re-localized to the lateral membranes of the expanding syncytium to re-distribute auxin for lateral expansion of the syncytium [15]. Of the 22 *ARF* genes in *Arabidopsis*, all except one (*ARF8*) were found to be up-regulated at the infection site, with distinct and overlapping spatiotemporal expression patterns, indicating *ARFs* may play different roles in various stages of syncytium formation [18]. At the same time, *miRNAs* target *ARF* mRNAs for degradation, and several *IAA* genes, which encode suppressors of ARF activities, are down-regulated at the infection site [19,20], further highlighting the sophisticated manipulation of auxin signaling for feeding site formation.

CNs also co-opt developmental programs of host cells through deployment of CLAVATA3 (CLV3)/ENDOSPERM SURROUNDING REGION (ESR) (CLE) peptide effectors [21–26]. In plants, CLEs are expressed as prepropeptides, and are then secreted into the apoplast where they are processed into mature 12–14 amino acid CLE peptides [27–30]. CLE peptides are perceived on the cell surface by leucine-rich repeat receptor-like kinases (LRR-RLKs) and

generally function in meristem cell homeostasis [31–34]. In nematodes, CLE effectors are injected through a stylet into the host cell cytoplasm in the form of a proprotein [35], and are then re-secreted through the endomembrane system into the apoplast [35–37], where they are processed into mature CLE peptides [29,30,38]. This re-secretion process not only allows CLE effectors to gain host-specific modifications that are critical for CLE activity [27,28,38–40], but it also gives the nematode access to the CLE signaling pathways of adjacent cells to prime these cells for syncytium incorporation [26,35,37].

Plant CLEs can be classified into A- and B-types, based on their ability to promote terminal differentiation of shoot and root apical meristem cells [41]. Both A- and B-type CLEs function in a CLE-Receptor-Like Kinase (RLK)-WUSCHEL(WUS)/ WUSCHEL-LIKE HOMEOBOX (WOX) paradigm to regulate stem cell activities. While A-type CLEs mainly functions in restricting plant shoot and root apical meristems (SAM/RAM) [42], B-type CLE, namely TRA-CHEARY ELEMENT DIFFERENTIATION INHIBITORY FACTOR (TDIF) [27], signals through TDIF receptor (TDR) to regulate a feed forward network to promote vascular (pro) cambial cell proliferation [43–48]. Interestingly, B-type CLE promotion of vascular stem cell proliferation can be enhanced by co-treatment of A-type CLEs like CLE6p, CLV3p, and CLE19p [41]. Although the molecular mechanism of synergistic action of A- and B-type CLEs is not clear, some *Arabidopsis* endogenous A-type CLEs do express in vascular tissue [49], and some showed activity in regulating vascular cell differentiation [50–56]. Remarkably, molecular mimics of both plant A- and B-type CLE peptides are found in CNs [21–26]. Both A- and B-type nematode CLE effectors signal through their respective receptors to promote cyst nematode parasitism [25,38,57–59], by, at least partially, activating *WOX4* gene expression [25].

Another important factor in regulating vascular stem cell homeostasis is the class III HOMEODOMAIN LEU-ZIPPER (HD-ZIP III) family of transcriptional factors [60]. The *Arabidopsis* HD-ZIP III family has five members, including *ATHB8*, *ATHB15/CORONA* (*CNA*), *PHABULOSA* (*PHB*), *PHAVOLUTA* (*PHV*), and *REVOLUTA* (*REV*). Expression of HD-ZIP IIIs is highly regulated by auxin biosynthesis and polar auxin transport [61,62], and at least one member, *ATHB8*, is directly regulated by AUXIN RESPONSE FACTOR 5 (ARF5)/ MONOPTEROS (MP) [63]. In the vasculature, HD-ZIP III genes show peak expression on the xylem side of vascular cambium [64], correlating with peak auxin concentrations [65,66]. Post-transcriptionally, HD-ZIP III levels are regulated by *microRNA 165/166* [67–73]. *MIR165/166* are activated in the endodermis, by stele-derived SHORT-ROOT (SHR) [67,74,75], and then diffuse back into the stele to establish a gradient of HD-ZIP III levels, where high HD-ZIP III level specifies metaxylem and low HD-ZIP III level specifies protoxylem [67,68]. Recently, Smetana *et al* (2019) established that the xylem-identity cell, which is promoted by high auxin and HD-ZIP III genes, functions as a stem cell organizer (SCO) to promote adjacent stem cell division and overall vascular patterning, reminiscent of the OC in SAM and QC in RAM [64].

Crosstalk between CLE and auxin signaling pathways regulates stem cell homeostasis and vascular patterning [48,76–79], and both CLE and auxin are known to be involved in CN-induced syncytium formation [10]. Nevertheless, it is not clear how these two phytohormones work together to promote syncytium formation. Here, we identified *ATHB8*, an HD-ZIP III family transcription factor, as a downstream factor of the HsCLE2 peptide effector, providing a potential intersection point for CLE and auxin signaling pathways in syncytium formation. We also demonstrated that *ATHB8*, along with other HD-ZIP III family genes, play an important role in beet cyst nematode parasitism of *Arabidopsis*. Distribution of *ATHB8* expression suggests that HD-ZIP III genes may function in both initial syncytial cell establishment and incorporation of adjacent cells for syncytium expansion, by promoting quiescence status of root cells.

## Results

### Gene expression profiles of the *clv1-101 clv2-101 rpk2-5* and wild type roots converge upon BCN infection

Our lab has previously reported that *Arabidopsis* receptor-like kinases CLV1, CLV2, and RPK2 perceive *Heterodera schachtii* (*H. schachtii*, BCN) CLE-like peptide effectors to promote parasitism [57,58], and that *WOX4* functions downstream of these CLE receptors to promote syncytium formation [25]. To isolate additional downstream CLE peptide signaling components involved in BCN parasitism, we profiled the transcriptome of BCN infection sites from both wild-type and *clv1-101 clv2-101 rpk2-5* triple (hereafter *clv* triple) CLE receptor mutant plants [25,58]. To filter out genes specifically responsive to nematode CLE effectors, we also included a group of samples treated with synthetic HsCLE2p effector (Fig 1A), a mimic of *Arabidopsis* CLE5/6p [24]. The treatment dose and timepoint was optimized to 5 μM HsCLE2p for 3 hours, using the expression of *WOX4* as the indicator (S1A Fig) [25]. The collected samples were verified for *WOX4* expression before proceeding to library construction (S1B Fig).

RNA sequencing yielded over 25 million reads from each sample (S1 Table). All marker genes were expressed as expected (S2A Fig). *clv1-101* and *clv2-101* are null alleles and *CLV1* and *CLV2* expression was barely detectable in the *clv* triple mutant; whereas *RPK2* showed high expression in the mutant due to the *rpk2-5* allele being a point mutation (S2D Fig; S. Sawa, personal communication). Expression of *WOX4* and *CLE41*, both known to function downstream of CLE signaling in BCN infection [25], were induced by HsCLE2p treatment in wild-type samples but not in the *clv* triple mutant (S2A Fig), indicating effective HsCLE2p treatment. PCA analysis showed that samples clustered as expected (S2B Fig), except that Rep 1 of HsCLE2p treated *clv* triple mutant clustered closer to wild-type samples rather than *clv* triple mutant samples (S2B Fig). Similarly, this sample also clustered with HsCLE2 treated wild-type samples in hierarchical clustering (S2C Fig). Despite the expression of all marker genes in this sample resembling the other two biological replicates of HsCLE2p treated *clv* triple mutant (S2A Fig), and confirmation of the *rpk2-5* mutation in this sample (S2D Fig), the sample was removed from the downstream analysis out of caution.

Differentially expressed genes (DEGs) were identified with the DEseq2 package [80]. In wild-type roots, 7,920 and 7,478 genes were up- or down-regulated, respectively, by BCN infection (padj < 0.05) (S3A Fig and S2 Dataset), and 2,550 and 2,672 genes were up- or down-regulated by HsCLE2p treatment (S3A Fig and S2 Dataset). In *clv* triple mutant roots, 7,706 and 7,383 genes were up- or down-regulated by BCN infection, and 1,178 and 1,530 genes were up- or down-regulated by HsCLE2 treatment (S3A Fig and S2 Dataset). Of the 7,920 up- and 7,478 down-regulated genes by BCN infection in the wild-type, 6,505 (82.1%) and 6,337 (84.7%) genes were also up- or down-regulated, respectively, by BCN infection in the *clv* triple mutant (S3B and S3D Fig). In comparison, with HsCLE2p treatment, only 30.98% (790 out of 2,550) of up-regulated and 36.5% (976 out of 2,672) of down-regulated genes showed the same regulatory trend between the *clv* triple mutant and the wild-type (S3C and S3E Fig). The higher percentage overlap of BCN-induced DEGs between the *clv* triple mutant and wild-type was also confirmed by Gene Ontology enrichment. The top 20 most enriched Biological Processes in BCN up- or down-regulated genes in wild type, were also enriched in the BCN up- or down-regulated DEGs in the *clv* triple mutant, often to a comparable enrichment level (S3F and S3H Fig and S3 Dataset). Whereas for HsCLE2p treatment, enriched GO terms in the *clv* triple mutant showed greater differences from that of the wild-type (S3G and S3I Fig and S3 Dataset). These results suggested that the syncytia that successfully establish in the *clv* triple mutant have a similar gene expression response compared to that of the wild-type, once they have circumvented the disruption of CLE signaling.

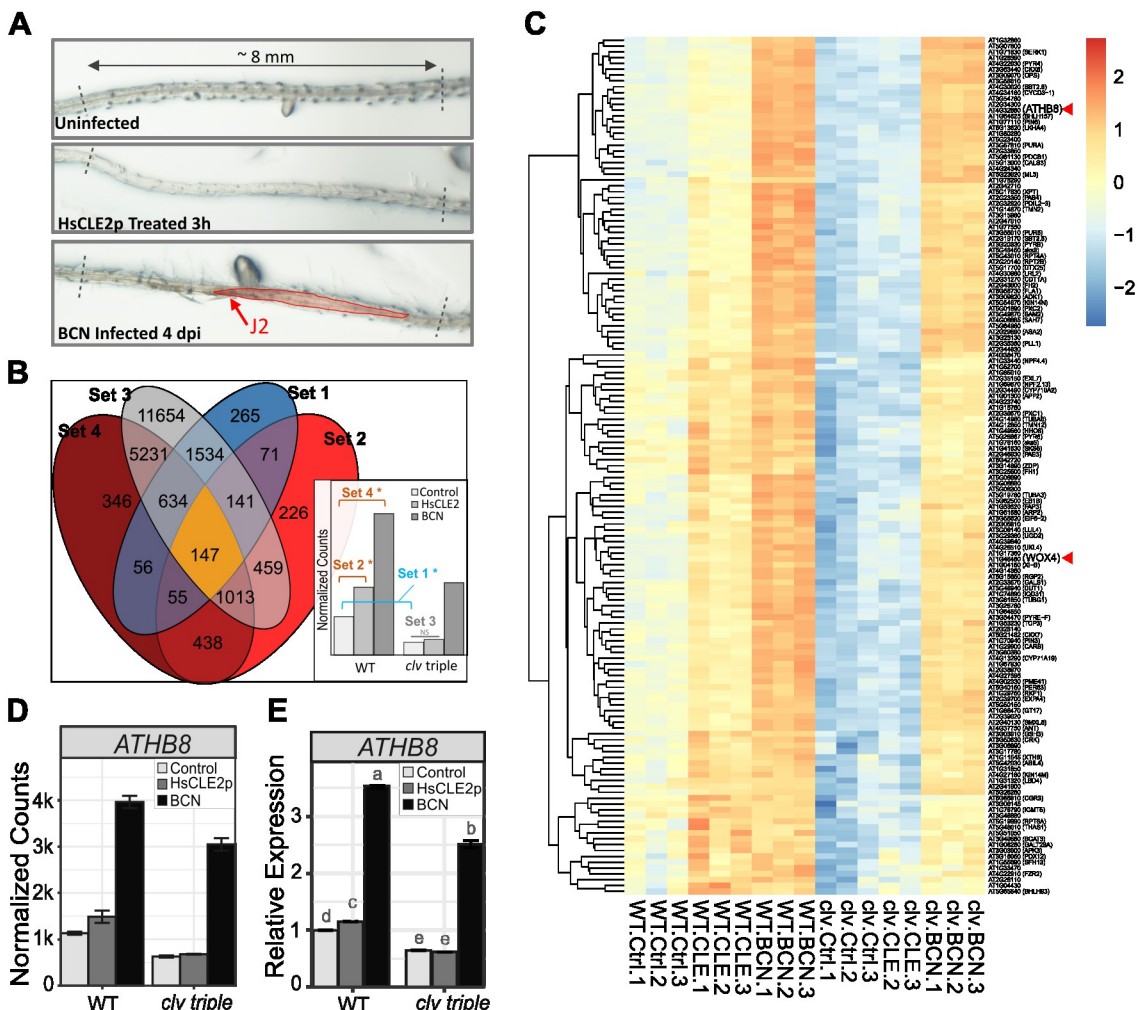

**Fig 1. Identification of potential genes downstream of CLE signaling upon BCN infection. A.** Example of excised root segments used for RNA sequencing. Approximately 8 mm long root segments spanning BCN infection sites and corresponding uninfected or HsCLE2p-treated root tissues were cut under a stereoscope and immediately frozen in liquid nitrogen for RNA isolation. **B.** Criteria used for filtering CLE downstream genes. Genes (147) that fit these four criteria were considered as CLE downstream genes, 1). down-regulated in the *clv* triple mutant compared to wild-type in control sample; 2). Up-regulated by HsCLE2p treatment in wild-type seedlings, but not in the *clv* triple mutant (3); 4). Up-regulated by BCN infection in wild-type seedlings. **C.** Heatmap showing expression profiles of the 147 candidate genes positively regulated by CLE signaling upon BCN infection. Expressions of *ATHB8* and *WOX4* are denoted with red arrows. **D.** Expression of *ATHB8* gene in the RNAseq dataset. **E.** qPCR validation of *ATHB8* gene expression in wild-type and *clv* triple mutant upon HsCLE2p treatment and BCN infection. Letter above each bar represents statistical groups of Tukey's HSD test following ANOVA analysis.

Interestingly, far less DEGs were identified between BCN-infected mutant and wild-type samples (S2 Dataset, *clv* BCN vs. WT_BCN, 2252 DEGs) compared to the number of DEGs found between uninfected *clv* triple mutant and wild type samples (S2 Dataset, *clv*_Ctrl.vs. WT_Ctrl, 5,388 DEGs) (S4A and S4B Fig). In contrast, HsCLE2p treatment resulted in an increased number of DEGs between mutant and wild-type (S2 Dataset, *clv*_HsCLE2 vs. WT_HsCLE2, 6440 DEGs) compared to untreated samples (S4C and S4D Fig). Again, these results suggested that BCN infection diminished gene expression differences between the transcriptome of the *clv* triple mutant and that of the wild-type. Consistently, BCN-infected wild-type and mutant samples clustered closer to each other compared to uninfected wild-type and mutant samples

(S2B and S2C Fig), indicating increased similarity between these two genetic backgrounds following nematode infection. These results suggest that BCN is capable of circumventing the disruption of CLE receptors, probably by signaling through alternative or parallel pathways, to induce syncytium formation. Such a compensatory mechanism is likely to obscure the identification of CLE downstream genes. Instead, HsCLE2p treatment provided a controlled approach to specifically activate CLE signaling pathways and identify downstream genes.

## Identification of downstream CLE signaling genes involved in BCN infection

We reasoned that genes promoted by CLE effectors via CLE receptors should have reduced expression levels in the mutant, due to attenuated endogenous CLE signaling, and should have an attenuated response to HsCLE2p treatment in the *clv* triple mutant. Moreover, these genes should also be promoted by BCN infection. Thus, we filtered genes that were, 1) down-regulated in the *clv* mutant compared to wild-type when uninfected, 2) up-regulated by HsCLE2p treatment in the wild-type root but not in the *clv* triple mutant, and 3) up-regulated by BCN infection. A total of 147 genes fit these criteria (Fig 1B and 1C and S4 Dataset). Using a similar approach, a set of 176 genes was identified as potential targets suppressed by the HsCLE2p effector (S5 Fig and S5 Dataset). For the time being, we primarily focused on the 147 HsCLE2p-promoted candidate genes. These 147 candidate genes were further divided into three tiers based on the level of induction in response to BCN infection, and their annotated molecular function (S4 Dataset). Genes having a log2 fold change larger than 2, or with regulatory related functions, were categorized as tier 1. To verify the expression pattern of these genes with qPCR, a new set of reference genes was selected and tested (S6B and S6C Fig), since previously published reference genes showed expression variations among our samples (S6A Fig). Thirty-two of the 38 tier 1 genes were selected for qPCR verification, of which most followed the expression pattern shown in the RNA sequencing data with few exceptions (S7 Fig).

## *ATHB8* is up-regulated at the periphery of the developing syncytium

One tier 1 gene, *ATHB8*, in particular, was identified as a strong candidate downstream CLE signaling gene involved in BCN infection. In the *clv* triple mutant, its response to HsCLE2p treatment was completely abolished (Fig 1D). Although it was still up-regulated by BCN infection in the mutant, the expression level did not reach that of wild-type (Fig 1D), indicating impaired CLE signaling compromised *ATHB8* activation upon BCN infection. This expression pattern was also verified by qPCR (Fig 1E). *ATHB8* belongs to the HD-ZIP III transcription factor family. Members of this five-member gene family in *Arabidopsis* are well known for their roles in vascular cell fate determination and vascular patterning [64,67,81]. In addition, all five members of the HD-ZIP III family were up-regulated by BCN infection (S8 Fig), indicating that these genes were likely to be involved in the differentiation of syncytial cells.

To explore the role of *ATHB8* in nematode parasitism, we first examined the activity of the *ATHB8* promoter in response to BCN infection using a *ProATHB8::GUS* transgene. In uninfected roots, the *ATHB8* promoter is only active in the vasculature and root apical meristem (RAM). It is highly expressed in RAM and gradually decreases through the division zone. In mature roots, the *ATHB8* promoter activity becomes slightly elevated again (S9 Fig), which is probably associated with its role in secondary vascular development [64,81]. Upon BCN infection, *ProATHB8::GUS* expression increased at the infection site compared to adjacent root tissue at early nematode developmental stages (Fig 2A). By the late J2 (2nd stage Juvenile) stage and on-wards, *ATHB8* was expressed at the periphery of the syncytium, rather than in the syncytium itself (Fig 2B–2E). However, *ATHB8* up-regulation becomes less prominent by the late

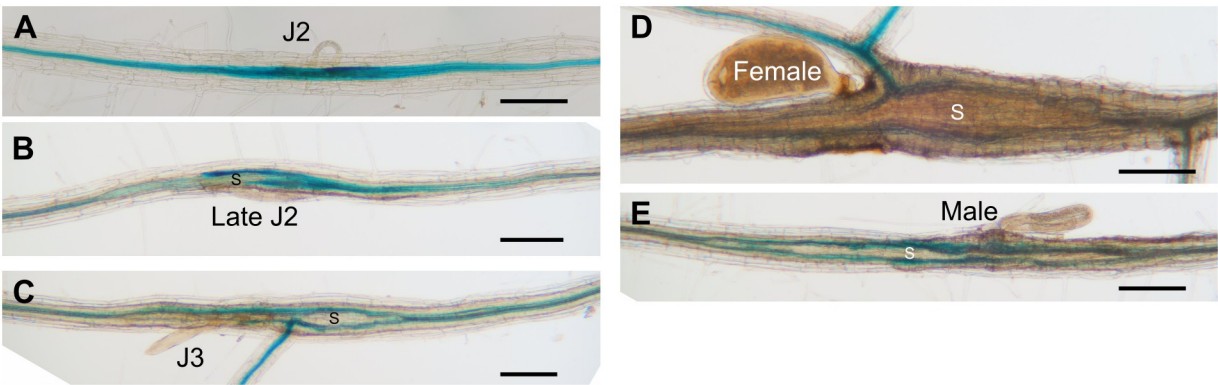

**Fig 2. Expression of *ProATHB8::GUS* in the BCN infection site.** *ProATHB8::GUS* expression at different stages of syncytium development in wild-type. bar = 200 μm. J2, second-stage juvenile; J3, third-stage juvenile; S, syncytium.

J2 or J3 stage compared to the early J2 stage (Fig 2A–2C). Once the nematode reached the adult female stage, *ATHB8* expression at the periphery of the syncytium was barely detectable (Fig 2D). Similarly, in male-associated syncytia, *ProATHB8::GUS* was also expressed in the periphery of syncytium but not in the syncytium itself (Fig 2E). We next examined a more detailed *ATHB8* expression at the early stage of syncytium formation using the *ProATHB8::4-xYFP* transgene. In uninfected seedlings, *ProATHB8::4xYFP* showed a similar expression pattern to that of *ProATHB8::GUS* (S10A Fig). An optical section near the root tip showed that *ATHB8* is expressed in the xylem axis in early vascular development (S10A Fig), consistent with previous reports [62,67]. In mature roots, *ATHB8* expression is predominantly located on the xylem side of vascular cambium [64]. Upon nematode infection, *ATHB8* expression level is clearly increased one day-post-inoculation (dpi) at the infection site, compared to adjacent root (Fig 3A). The YFP signal continues to increase up to about 4 dpi before it starts to decrease (Fig 3A). We further quantified the YFP signal intensity at multiple independent infection sites, all of which showed a clear increase of YFP level at infection site compared to an adjacent root segment (Figs 3A and S11A–S11G). Optical sections of a five-day old syncytium showed that the YFP signal is located at the periphery of the syncytium (S10B Fig), consistent with what was observed with the *ProATHB8::GUS* transgene (Fig 2B).

## *MIR165a* expression is down-regulated at the BCN infection site

*ATHB8*, as a member of the HD-ZIP III family of genes, is post-transcriptionally regulated by the *MIR165/166* family of microRNAs. *Arabidopsis* encodes two *MIR165* genes and seven *MIR166* genes, of which *MIR165a*, *MIR166a*, and *MIR166b* are expressed in roots, and only *MIR165a* and *MIR166b* promoters drive detectable GFP expression [67,82]. *MIR165/166* genes are expressed in the endodermis cells and move into the stele to suppress HD-ZIP III gene expression [67]. To examine the response of *MIR165/6* to BCN infection, two transgenic lines, *ProMIR165a::GFP* and *ProMIR166b::GFP*, were used to track their expression in the infection site. In uninfected roots, *ProMIR165a::GFP* is expressed in the endodermal cells in both the root tip and mature root (S12A Fig) [67]. Upon BCN infection, expression of *ProMIR165a::GFP* signal is gradually reduced at the infection site compared to the adjacent root segment (Figs 3B and S13A–S13G); whereas for *MIR166b*, GFP signal was only detected in the endodermal cells of root tips in both uninfected and infected plants (S12B Fig). No signal was detected in the infection site up to five days post infection (S12C and S12D Fig). Furthermore, in the

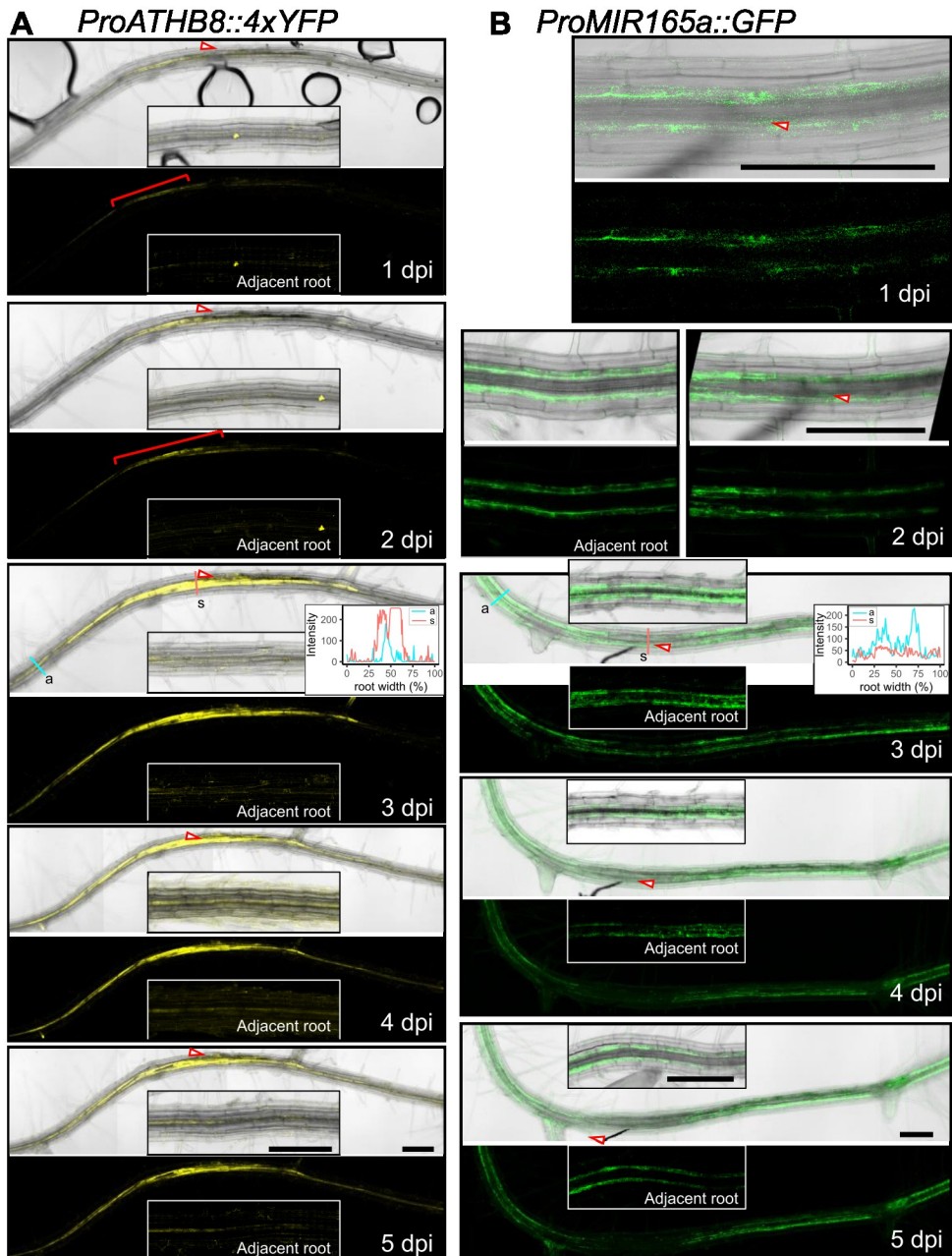

**Fig 3. Expression of *ProATHB8::4xYFP* and *ProMIR165a::GFP* at BCN infection sites in *Arabidopsis* roots. A.** Continuous monitoring of *ProATHB8::4xYFP* expression in early stages of BCN infection. Red brackets indicate increased YFP signal at early stages of BCN infection. Insets represent YFP signal of root segment about 850 μm away from the infection site. YFP signal intensity at 3 dpi was quantified at the syncytium (s, red line) or adjacent site (a, cyan line). **B.** Continuous monitoring of *ProMIR165a::GFP* in early stages of BCN-infected roots. Left panel of 2 dpi, and insets of 3, 4, and 5 dpi panels represent GFP signal of root segment adjacent to the infection site. GFP signal intensity at 3 dpi was quantified at the syncytium (s, red line) and adjacent site (a, cyan line). Red arrowhead, position of nematode head. bar = 200 μm.

RNAseq dataset, *AGO1* (*AT1G48410*), a gene encoding an ARGONAUTE protein that recruits MIR165/6 for target mRNA cleavage [83], is downregulated; while *AGO10* (AT5G43810), which encodes an ARGONAUTE that specifically sequesters *MIR165/6* for degradation [84–86], is up-regulated (S14 Fig). These results point to a reduced *MIR165/6* level and activity at the BCN infection site.

To test if the MIR165/6 activity is indeed reduced, the *U2::MIR165/6-GFP MIR165/6 sensor* line, which harbors a MIR165/6 target sequence at the 3'UTR of ER-GFP construct [67], was used to monitor the MIR165/6 activity at the early stage of BCN infection. In this system, high *MIR165/6* activity is represented by low GFP expression [67]. In uninfected roots, high GFP expression was observed in the vasculature, representing low levels of *MIR165/6* activity in corresponding tissue (S15A Fig). After BCN infection, we did not observe a clear increase in GFP signal, due to the preexisting high level of GFP expression (low MIR165/6) (S15B Fig); however, the GFP expression domain was slightly expanded at the infection site (S15B Fig), probably due to reduced MIR165 expression in the endodermis cells (Figs 3B and S13).

## Loss of function of *ATHB8* and *ATHB15/CNA* does not affect BCN infection

To test if *ATHB8* plays any role in syncytium formation, we obtained single and double knock-out mutants of *ATHB8* and its closest homology *ATHB15/CNA*, *athb8-11* and *cna-2* respectively [67,87]. In our growth conditions, young seedlings of *athb8-11*, *cna-2*, and *athb8-11 cna-2* mutants all showed similar root and shoot growth and development compared to that of the wild-type control Col *er-2* (Fig 4A–4C). Comparable numbers of BCN juveniles were able to penetrate into the roots of these mutants and wild-type plants (Figs 4D and S16A). The number of adult females that developed on these mutants' roots, and size of syncytium associated with single female were also comparable to that of wild-type plants (Figs 4E, 4F, S16B and S16C). These results indicated that knocking out *ATHB8* and *ATHB15/CNA* genes does not affect syncytium formation and syncytium expansion in the *Arabidopsis* root, probably due to high redundancy of HD-ZIP III family genes in mediating vascular cell differentiation, and that all five members were up-regulated at the BCN infection site [67,87](S8 Fig).

## Knock down of the HD-ZIP III gene family reduced BCN infection

To overcome functional redundancy among HD-ZIP III genes, higher order mutants would be needed for BCN infection assays. However, vascular development defects in these high order mutants, especially quadruple and quintuple HD-ZIP III mutants, results in severe root growth phenotypes, making them unsuitable for assessing BCN infection phenotypes [64,67,87]. To further evaluate the role of HD-ZIP III genes in cyst nematode infection, we adapted the estradiol-based inducible gene expression system to the cyst nematode infection assay [88]. Two *MIR165a*-inducible expression lines, *Pro35S::XVE>>MIR165a* and *ProCRE1::XVE>>MIR165a* (hereafter designated as *Pro35SiMIR165a* and *ProCRE1iMIR165a*) [64,89], were used to knock down HD-ZIP III gene expression at the time of nematode infection. Upon 48 hours of induction by 5 μM 17-beta-estradiol, expression of all members of the HD-ZIP III gene family in the *Pro35SiMIR165a* line were significantly suppressed (Fig 5A). Suppression of gene expression in *ProCRE1iMIR165a* were less severe compared to that of the *Pro35SiMIR165a* line, yet expression of all members of HD-ZIP IIIs except for *REV* were significantly decreased (Fig 5A). Increasing estradiol concentration to 10 μM, or prolonging the incubation time to 72 hours, did not further reduce target gene expression (Fig 5A). In fact, when estradiol induction was extended beyond three days, the expression levels of all HD-ZIP III genes gradually recovered, highlighting the transient effect of estradiol induction (Fig 5B).

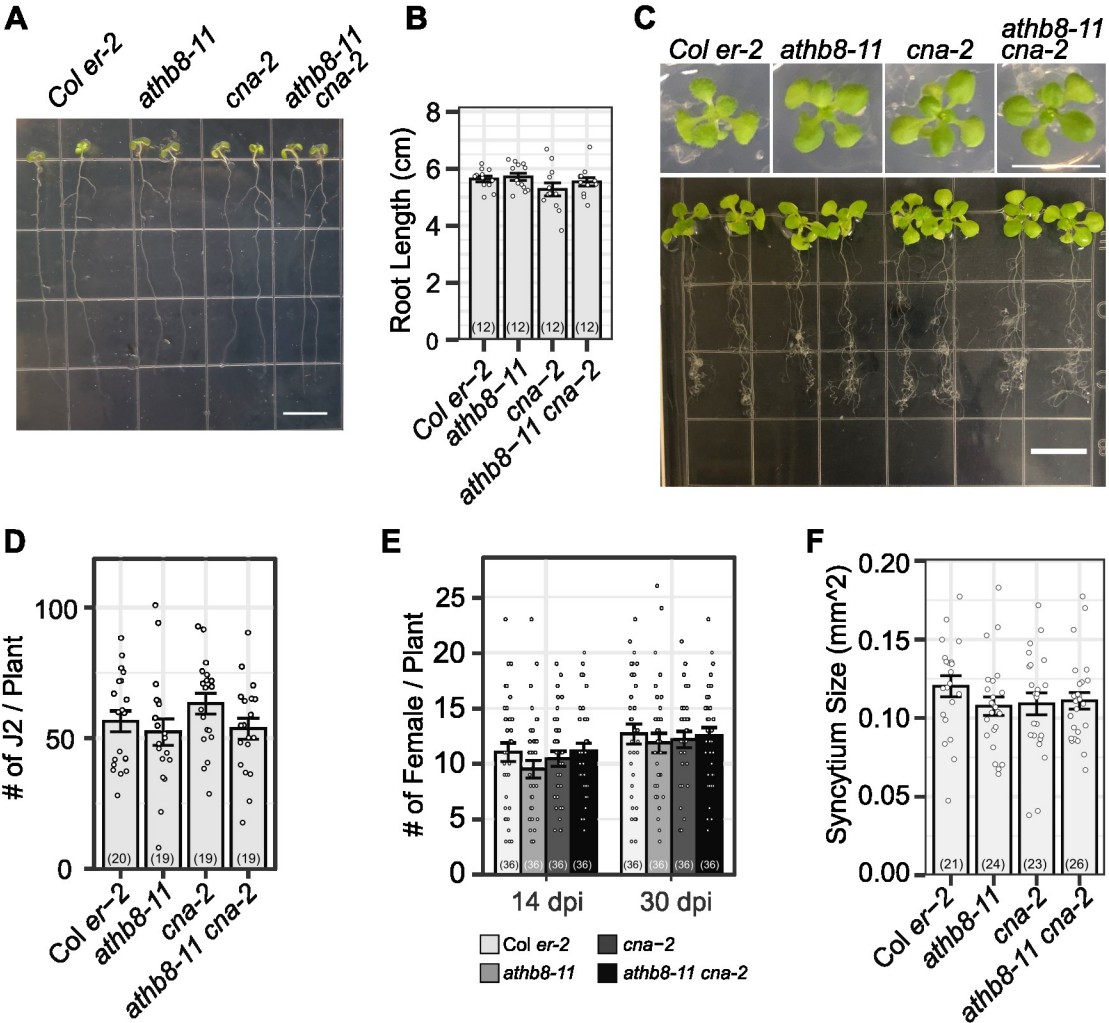

**Fig 4. Loss of function of *ATHB8* and its closest homologue *ATHB15/CNA* do not affect BCN infection or syncytia development in *Arabidopsis*. A.** 7-day-old seedlings of *athb8-11*, *cna-2*, and *athb8-11 can-2* grown on a vertical plate. bar = 1 cm. B. Measurement of root length shown in (A). **C.** 14-day-old seedlings grown in 12-well plates. Top panel, top view of representative seedlings. Bottom panel, root system of representative 14-day-old seedlings grown in 12-well plates and scooped out for photograph. bar = 1 cm. **D.** BCN penetration rate on the tested mutants, counted at 3–5 days post inoculation. **E.** Number of adult females developed on the tested mutants. **F.** Syncytium size developed on the tested mutants. All bar graphs represent mean ± SE. Dots represent each individual measurement. Samples sizes were shown in parentheses in each bar. Statistical tests were performed with Wald test following generalized linear mixed-effect model (for penetration and infection data) or linear mixed-effect model (for syncytium size). Data were repeated twice with similar results (S16 Fig).

This expression profile, however, is advantageous for nematode infection assays, as the disruption of the target gene can be timed with nematode infection, and any adverse effects of target gene knock-down on plant development can be largely minimized (Fig 5B). Based on this information, 5 μM estradiol was used for *MIR165a* induction on 12-day-old seedlings and BCN was inoculated 48 hours after induction [88]. Estradiol and DMSO treated *Pro35SiGUS* lines showed similar penetration and infection rate, as well as female associated syncytium size (Fig 5C–5E). These numbers were also comparable to nematode infection on wild-type plants without any treatment (Fig 4D and 4E), consistent with our previous finding that estradiol treatment did not affect BCN activity [88]. In either *Pro35SiMIR165a* or *ProCRE1iMIR165a*

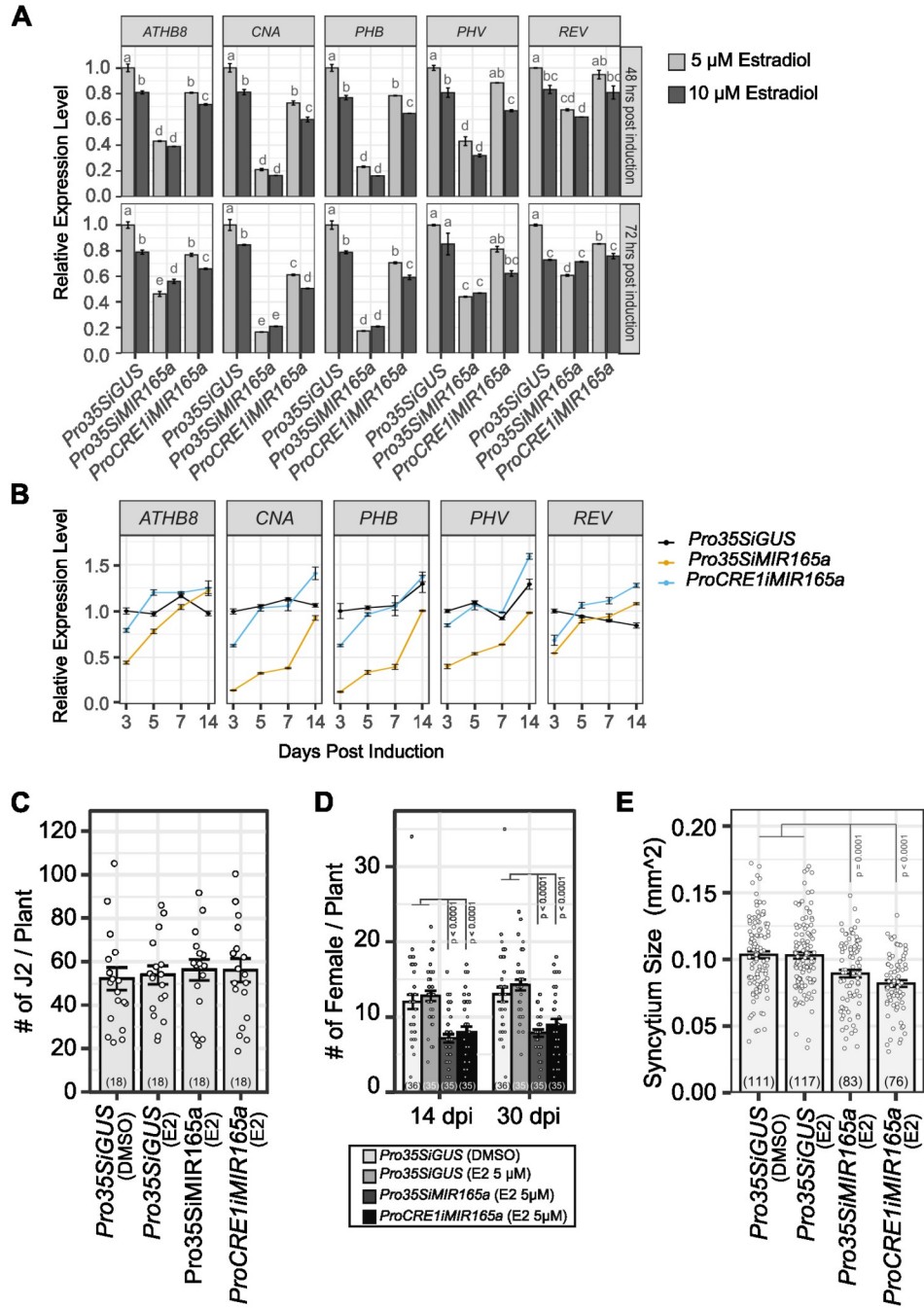

**Fig 5. Inducible expression of *MIR165a* suppresses BCN infection in *Arabidopsis*. A**. Induction of *MIR165a* suppressed expression of HD-ZIP III TFs in 12-well plates. Top panel, 48 hours post-induction. Bottom panel, 72 hours post-induction. Letter above each bar represents statistical group of Tukey's HSD test following ANOVA analysis. **B.** Expression of HD-ZIP III TFs 3–14 days post induction of *MIR165a* with 5 µM estradiol (E2). Effect of *MIR165a* induction attenuates over time.**C.** Suppression of HD-ZIP III TFs did not affect nematode penetration into the root. Penetration rate counted at 5 dpi. **D.** Suppression of HD-ZIP III TFs suppressed BCN female development on *Arabidopsis* roots. **E.** Suppression of HD-ZIP III TFs resulted in smaller syncytia size. Bar graph represents mean ± SE. Dots represent each individual measurement. Sample sizes were shown in parentheses in each bar. Statistical tests were performed with the Wald test following a generalized linear mixed-effect model (for penetration and infection data) or linear mixed-effect model (for syncytium size). Infection assays were repeated four times with similar results (S17 Fig). Four reps of syncytium size data were unified using biological replications as a block effect to build the linear mixed-effect model. Only statistically significant results were labeled on the graph. E2, estradiol.

line, nematode penetration rates were not significantly affected by estradiol treatment, counted at five days post inoculation (Figs 5C and S17A). Whereas the development of adult females on both inducible lines was strongly suppressed (Figs 5D and S17B), indicating that HD-ZIP III TFs do play essential roles in syncytium establishment. In addition, average syncytium size in both *Pro35SiMIR165a* and *ProCRE1iMIR165a* lines was also significantly reduced compared to the control in four biological replicates (Figs 5E and S17C). Interestingly, although the *ProCRE1iMIR165a* line only had a slight decrease and fast recovery of HD-ZIP III gene expression (Fig 5A and 5B), reductions in both female development and syncytium size in this line were comparable to that of the *Pro35SiMIR165a* line (Figs 5D, 5E, S17B and S17C), suggesting that reduced HD-ZIP III expression in the stele is sufficient to suppress syncytium formation and expansion in BCN infection.

To determine if HD-ZIP III genes other than *ATHB8* and its closest homologue *ATHB15/CNA* contribute to reduced infection rate, we conducted infection assays of other HD ZIP III mutants, including *phb-13*, *phv-11*, *rev-6*, and a *phb-13 phv-11 cna-2* triple mutant. In our growth conditions, these mutants did not show obvious root growth defects (S18A and S18B Fig). We also did not observe any differences in nematode penetration (S18C and S18F Fig), infection (S18D and S18G Fig), or syncytium size (S18E and S18H Fig) in these mutants compared to that of wild-type. These results further confirmed that all HD-ZIP III genes including *ATHB8* contribute to syncytium formation.

## Overexpression of *ATHB8* does not increase susceptibility of *Arabidopsis* to BCN

Next, we tested if overexpression of *ATHB8* would result in cyst nematode hyper-susceptibility. *ATHB8* overexpression lines were created by crossing *ProANT::XVE>>ATHB8d-YFP* (*ProANTiATHB8d-YFP*), which expresses *MIR165* resistant *ATHB8* (*ATHB8d*) using the *ANT* promoter [64], with *Pro35S::XVE>>GUS* (*Pro35SiGUS*) and *ProG1090::XVE>>GUS* (*ProG1090iGUS*) lines. F1 of these crosses, designated as *Pro35SiATHB8d-YFP* and *ProG1090iATHB8d-YFP* respectively, contain constructs from both parents thus the *ATHB8d-YFP* transgene can be activated by both the *ANT* promoter and *35S/G1090* promoters, resulting in ubiquitous overexpression. Indeed, in the root of F1 plants, *ATHB8d-YFP* expression domains were much expanded compared to that of the parent line *ProANTiATHB8d-YFP* (Fig 6A). qPCR showed that in all lines, expression level of *ATHB8* was highly induced three days post induction, up to 70 times higher in the *ProG1090iATHB8-YFP* line, and 10 times higher in *Pro35SiATHB8d-YFP* and *ProANTiATHB8d-YFP* lines (Fig 6B). Consistent with previous observations (Fig 5B), the levels of induced gene expression declined over time (Fig 6B). Expression of other HD-ZIP III TF family members was not substantially affected by *ATHB8d* overexpression (Fig 6B). In all tested lines, overexpression of *ATHB8d* did not affect penetration of nematodes into the root (Figs 6C and S19A) or average syncytia size induced by nematodes (Figs 6E and S19C). In *35S* and *ANT* promoter driven lines, which have relatively low overexpression levels, female development was not significantly affected (Figs 6D and S19B). However, in the *ProG1090iATHB8d-YFP* line, nematode female development was strongly suppressed (Figs 6D and S19B). This distinct phenotypic difference in female development between *G1090* driven and *35S/ANT* driven lines may be due to the extremely high *ATHB8d-YFP* expression level and/or expanded expression domain of the *G1090* promoter compared to the other two lines (Fig 6A and 6B). Induction of *ProG1090iATHB8d-YFP* in younger seedlings (5-day old) for seven days caused severe developmental defects in both roots and shoots, regardless of BCN infection (S20 Fig), while *ProANTiATHB8d-YFP* and *Pro35SiATHB8d-YFP* did not show a clear phenotype (S20 Fig). However, developmental defects of the

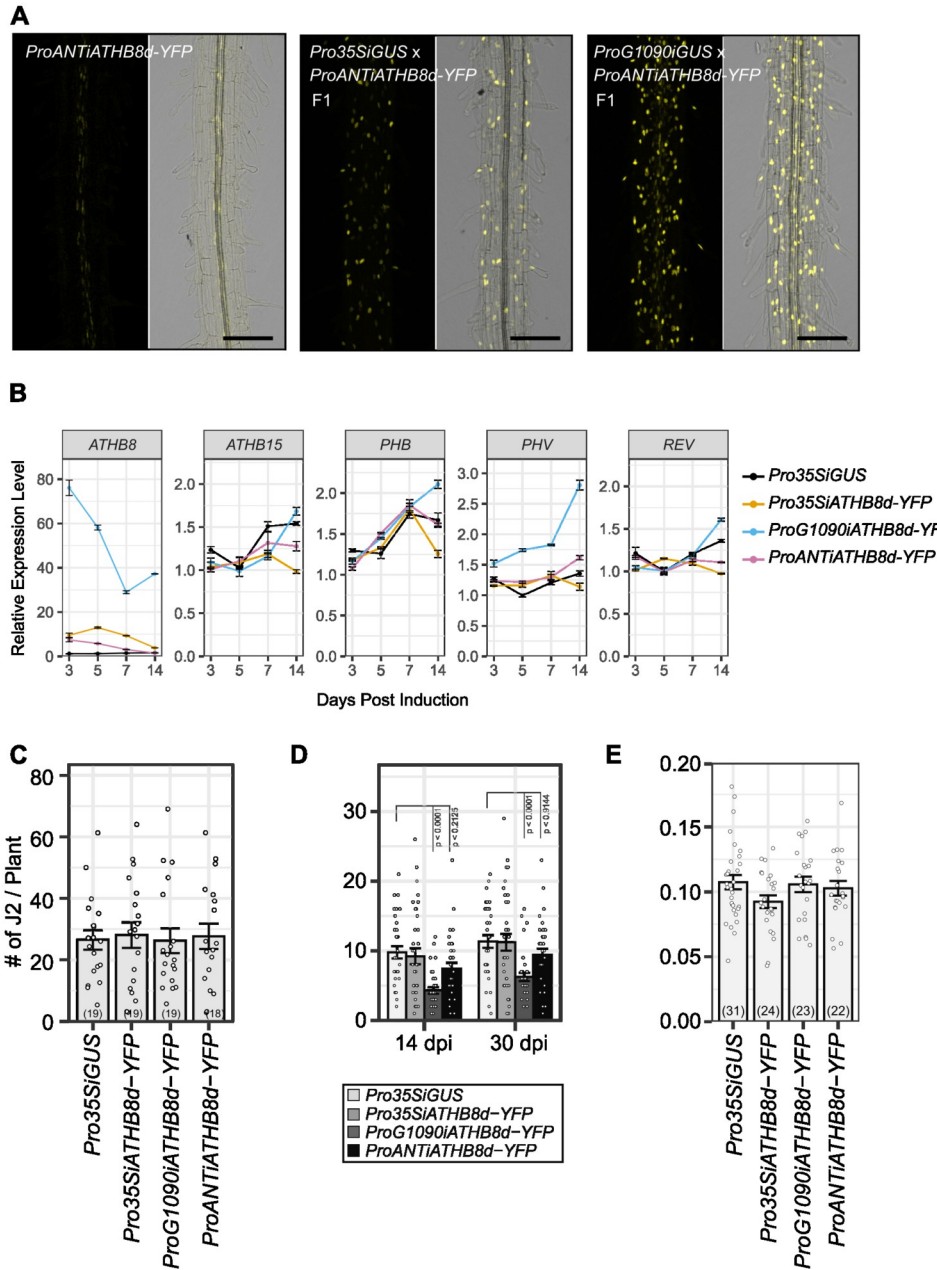

**Fig 6. Effect of overexpressing *ATHB8* on cyst nematode infection in *Arabidopsis*. A**. Expanded ATHB8d-YFP expression domain after introduction to *Pro35SiGUS* and *ProG1090iGUS* lines. 5-day-old seedlings were induced with 5 μM estradiol for 24 hours before imaging. bar = 100 μm. **B.** Expression of HD-ZIP III genes in 12-well plates after estradiol (5 μM) induction. **C-E.** Effect of *ATHB8d* overexpression on BCN penetration (**C**), female development (**D**), and syncytia size (**E**) in *Arabidopsis* root. Bar graph represents mean ± SE. Dots represent each individual measurement. Sample sizes were shown in parentheses in each bar. Statistical tests were performed with the Wald test following a generalized linear mixed-effect model (for penetration and infection data) or a linear mixed-effect model (for syncytium size). Data shown in C–E were repeated twice with similar results (S19 Fig).

*ProG1090iATHB8d-YFP* line were not obvious for seedlings used in 12-well plate infection assays (S21 Fig), where seedlings were much older (12-day) and had more developed root systems that are more resilient to *ATHB8d* overexpression (S20 Fig). Nevertheless, the defects in root development were unlikely to directly contribute to reduced female development, as the root system has not been significantly affected by *ATHB8d-YFP* overexpression at the point of BCN inoculation, and comparable numbers of nematodes were able to penetrate into the root system in all lines (Figs 6C and S19A). These results suggested that overexpression of *ATHB8* is not sufficient to induce BCN hyper-susceptibility. Instead, the balance of stem cell organizer cell specification and its further differentiation might be important for syncytial cell transition.

### Auxin level and *ATHB8* expression are elevated in neighboring cells of the developing syncytium

Considering the finding that *ATHB8* expression is upregulated at the periphery of syncytium (Figs 2 and S10B), coupled with prior studies demonstrating *ATHB8* can be directly regulated by auxin through ARF5/MP, an auxin responsive factor essential for vascular development [63], and that auxin was proposed to function in conditioning syncytium periphery cells for syncytial integration [14,15], we reasoned that *ATHB8* may function downstream of auxin in conditioning cells for incorporation into the syncytium. To explore this hypothesis, we carried out a detailed analysis of the spatial distribution of auxin response and *ATHB8* promoter activities at early stages of syncytium formation. Indeed, at 24 hpi (hours post inoculation), both *ProATHB8::4xYFP* and *DR5::4xYFP* were seen in the developing syncytia (Figs 7A, 7B, S22A and S22B). By this stage, the syncytia were already well developed with extensive cell wall modifications and fused cytoplasm (Figs 7A, 7B, S22A and S22B), and higher YFP signal was frequently observed in cells neighboring syncytial cells (Figs 7A, 7B, S22A and S22B). Nevertheless, to determine if auxin level or *ATHB8* expression peaked earlier than this is technically challenging, as it is difficult to determine if the nematode has begun feeding yet, or pinpoint the initial syncytial cell without visible cell wall modification or a reliable syncytium specific marker. In both *ProATHB8::4xYFP* and *DR5::4xYFP* lines, the YFP signal was also observed in pericycle cells incorporated into the syncytium (S22B Fig), or pericycle cells near the syncytium (Fig 7B). These two transgenes are typically not expressed in pericycle cells of uninfected roots, except where lateral roots initiate [90]. At 3 dpi, YFP signal in the syncytium was barely detectable in the syncytial cell itself, but was observed primarily in cells neighboring the developing syncytium (Figs 8A–8C and S23A–S23C). These results showed that indeed, auxin and *ATHB8* levels first accumulate in the early-stage syncytial cells and then shifts to adjacent cells of the developing syncytium [15], implying that auxin signaling and *ATHB8*, along with other HD-ZIP III genes, may function in promoting the transition of a host cell into a syncytial cell.

## Discussion

### Gene transcription profiles of CLE receptor mutant and wild-type roots converge upon cyst nematode infection

Cyst nematode CLE-like peptide effectors were the first stylet-secreted effectors found to mimic host peptide hormones to modulate plant developmental programs for parasitic success [22–24]. Impairing CLE receptors in host plants attenuates the CLE signaling pathway and increases host resistance to cyst nematode infection [25,40,57–59]. *Arabidopsis* CLE receptors CLV1, CLV2, and RPK2, and their orthologs in soybean and potato, play an important role in mediating CN infection [25,40,58]. Simultaneously knocking out or knocking down the three

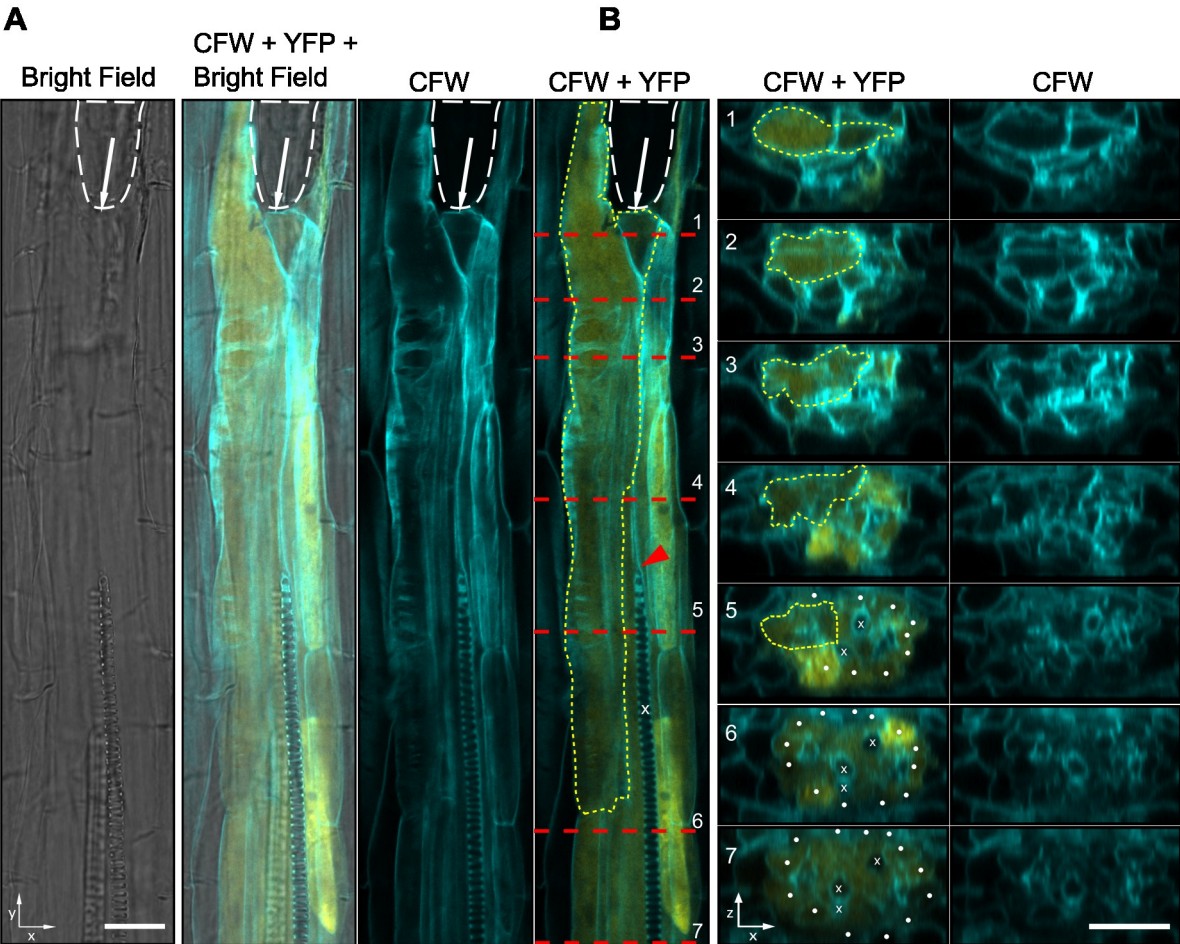

**Fig 7. Expression of *ProATHB8::4xYFP* in the BCN induced syncytium at 1 dpi. A**. A confocal optical section of developing syncytium with *ProATHB8::4xYFP* expression. **B**. Optical cross section of positions shown in (**A**). White dashed line, outline of nematode head. White arrow, position of the stylet. Red dashed line, positions of cross section shown in (**B**). Yellow dashed line, outline of the syncytium. x, xylem cells. Red arrowhead, disrupted xylem differentiation. White dot, pericycle cells. CFW, calcofluor white staining of cell wall. YFP, yellow fluorescent signal. bar = 20 μm.

receptors results in lower nematode infection rate and smaller syncytia size [25,40,58]. Like in other classical CLE-RLK-WUS/WOX signaling modules, the *WOX4* gene, which functions in maintaining vascular cambial activity [43,45,48,91], is up-regulated by nematode infection through A- and B-type CLE receptors [25]. Nevertheless, knocking out *WOX4* does not yield similar levels of BCN resistance as the *clv1-101 clv2-101 rpk2-5* triple CLE receptor mutant, indicating other CLE downstream factors are at play during BCN infection [25]. Isolating such downstream factors by directly comparing BCN-infected wild-type and triple CLE receptor mutant transcriptomes proved difficult, as BCN infection promoted transcriptome convergence between the mutant and wild-type (S2B, S2C, S3B, S3D, S3F and S3H Figs). These results suggested that once BCN overcame the initial barrier of CLE signaling impairment, they were capable of inducing an attenuated, yet similar gene expression profile sufficient for syncytium establishment and maintenance, albeit with some size restriction that may compromise feeding and lead to developmental delays [57,58]. This convergence of gene expression profiles of mutant and WT roots responding to BCN infection also highlights the robustness

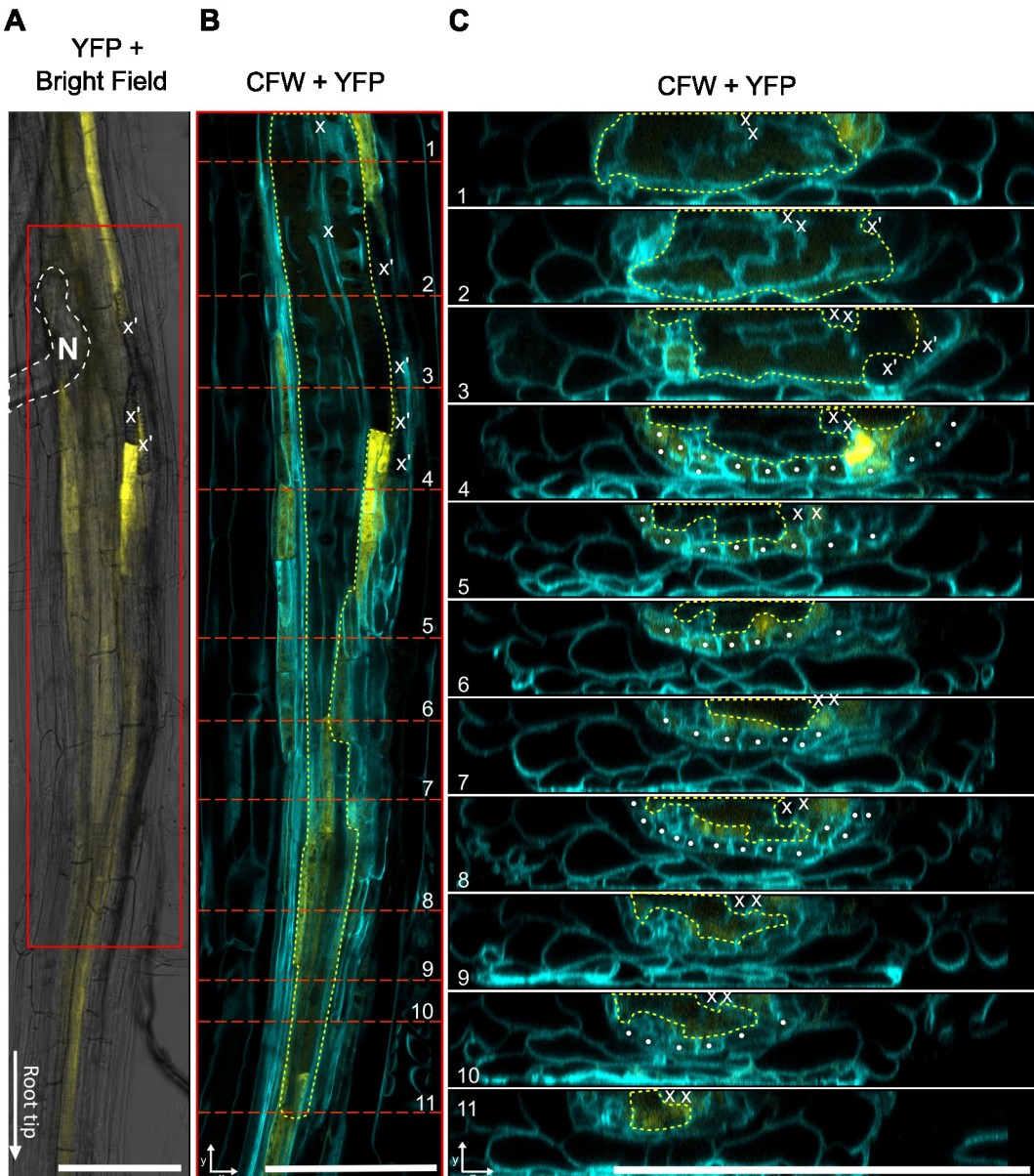

**Fig 8. Distribution of *ProATHB8::4xYFP* activity at BCN infection site at 3 dpi.** White dashed line, outline of nematode. Red box, zoomed in portion in (B). Red dashed lines in (B), position of optical cross sections shown in (C). Yellow dashed line, outline of the syncytium. x, xylem cells. x', ectopic xylem cells. White dot, pericycle cells. CFW, calcofluor white staining of cell wall. YFP, yellow fluorescent signal. bar = 100 μm.

of nematode effectors in co-opting plant developmental programs. The indifference of syncytia transcriptomes between the triple CLE receptor mutant and wild-type could be due to secretion of a mixture of A- and B- type CLE peptide mimics [24,25], that could signal through multiple CLE receptors and function redundantly or in parallel to bypass these impaired CLE receptors [42]. Moreover, modulation of other signaling pathways, like auxin and cytokinin, by other nematode effectors, could also compensate for the lack of these CLE receptors, given that these signaling pathways are highly intertwined with CLE signaling in regulating vascular

cambium activity [92–94]. For example, *WOX4*, the infamous gene in regulating vascular cambium activity downstream of CLE signaling, is also responsive to auxin and cytokinin, both of which function in modulating a complex transcriptional network to regulate vascular cambium activity [46,91,95–97]. During cyst nematode infection, an overdrive of these hormone signaling pathways could contribute to the up-regulation of CLE downstream genes like *WOX4*, thereby masking effects of attenuated CLE signaling on their expression.

HsCLE2p treatment provided a more controlled and effective way to isolate CLE downstream genes. Genes identified in this way (Figs 1B, 1C and S5) are likely to be directly regulated by CLE peptide signaling, rather than a secondary response, as they responded relatively quickly (3 hours) to a relatively low dose (5 μM) of HsCLE2p treatment, on par with the response of the *WOX4* gene (S1A Fig). The list of genes positively regulated by CLE signaling was particularly interesting to us. Not only because *WOX4*, the only known marker gene for CLE signaling in vascular development and cyst nematode infection, was included in the list, but because this list also included a number of genes known to be involved in vascular development or cell fate determination, such as *BAM2*, *SERK1*, *PXC1/2*, *ANT*, *CYCD3;1*, *LBD4*, *OPS*, and *ATHB8* (S4 Dataset) [46,64,81,98–104], with a few of these genes reported to be regulated by CLE signaling [41,43,46] or involved in syncytium formation [10, 15, 105].

## *ATHB8* gene expression is regulated by CLE peptide signaling and BCN infection

*ATHB8* belongs to the HD-ZIP III family of transcription factors, which play essential roles in vascular development [60]. *ATHB8* has been shown to be up-regulated by 10 μM CLE41p/ TDIF (B-type) within one day of treatment, and co-application of CLE6p (A-type) with CLE41p accelerated *ATHB8* up-regulation to 3 hours [41]. Here, we showed that application of 5 μM HsCLE2p, a peptide effector mimic of CLE6p [24], induced significant up-regulation of *ATHB8* within 3 hours of treatment (Fig 1D and 1E), a timeline on par with co-treatment of A- and B- type CLEs [41]. Like the *tdr-1* mutant, the *clv1-101 clv2-101 rpk2-5* mutant eliminated the response of *ATHB8* to corresponding CLE peptide treatments [25] (Fig 1D and 1E), suggesting corresponding receptors are required for A- or B-type CLE-induced *ATHB8* expression at this time point. It is possible that HsCLE2p/CLE6p promotes *ATHB8* expression by up-regulating *CLE41* gene expression (S2A Fig), which in turn signals through the TDR-BIN2/BIL1 module to activate ARF5/MP and its homologues to promote *ATHB8* gene expression [63,79,106]. Nevertheless, it is also likely that HsCLE2p/CLE6p has its own pathway to regulate *ATHB8* expression, given that co-application of HsCLE2p/CLE6p greatly enhanced CLE41p/TDIF-induced *ATHB8* expression [41].

*ATHB8* expression was also strongly up-regulated by BCN infection (Figs 1D–1E, 2A–2E and 3A). In early syncytial cells, *ATHB8* promoter activity was elevated compared to adjacent cells (Figs 2A, 7A and 7B), but as the syncytium expanded expression became more prominent in cells neighboring the syncytium rather than in the syncytium itself (Figs 2B, 2C, 8B, 8C and S10B). This shift of the *ATHB8* active domain could be a result of re-secretion of CN CLE effectors that activated CLE signaling pathways in syncytium neighboring cells [35–37]. In additional, lateral re-distribution of auxin could also play important roles, as *ATHB8* is also directly regulated by auxin [14,15]. Accumulation of *ATHB8* transcript could also be augmented by suppression of *MIR165/166* expression at the infection site (Figs 3B and S13). *MIR165/166* gene activation requires transcription factor SHOOT-ROOT (SHR) to move from stele cells to the endodermis layer through plasmodesmata [67,74,75,107–110]. Symplastic isolation of the syncytial cell could restrict SHR movement and prevent *MIR165/166* gene expression from being activated [8,111–117]. Furthermore, the *AGO1* gene, which encodes an

ARGONAUTE that recruits microRNAs for mRNA cleavage [83], was down-regulated (S14 Fig); while the gene encoding AGO10, which specially sequesters *MIR165/166* and promotes their degradation [84–86], was up-regulated (S14 Fig). Collectively, these mechanisms could regulate *ATHB8* expression level in and around syncytium and result in high *ATHB8* transcript accumulation (Fig 1D and 1E).

Co-existence of multiple mechanisms in regulating *ATHB8* expression ensures that disrupting CLE signaling alone will not completely abolish BCN-induced *ATHB8* up-regulation, like in the situation of HsCLE2p treatment (Fig 1D and 1E). Thus, it is not surprising that *ATHB8* was also highly up-regulated at the BCN infection site in the *clv* triple mutant (Fig 1D and 1E). Nevertheless, the expression level of *ATHB8* in the mutant did not reach to the same level as the wild-type infection sites (Fig 1D and 1E), suggesting that CLE signaling is indeed required for optimal *ATHB8* expression during syncytium formation.

## HD-ZIP III transcription factors function in syncytium establishment

*ATHB8*, along with other HD-ZIP III family members, promote xylem differentiation during vascular development in a dose-dependent manner [67,81,87]. Only when all five HD-ZIP III genes are knocked out is xylem differentiation completely blocked, indicating all five members contribute to xylem differentiation [67]. Functional redundancy among HD-ZIP III members is also present in syncytium formation, as knocking out individual members and/or their closest homologue was not sufficient to reduce the rate of BCN infection or average syncytium size (Figs 4E, 4F, S16B, S16C; S18D, S18E, S18G and S18H). Consequently, knocking down HD-ZIP III genes by inducible ectopic expression of *MIR165a* dramatically reduced the rate of successful BCN infection without affecting nematode penetration (Fig 5C and 5D). It is worth noting that expression of all five HD-ZIP III genes were still detectable after estradiol induction (Fig 5A), and that their expression levels gradually recovered over time after BCN inoculation (Fig 5B). If it were possible to knock out the activity of these genes locally in response to infection, syncytium establishment might be completely suppressed, given the indispensable role of HD-ZIP III genes in vascular patterning [67]. Interestingly, despite the fact that the *Pro-CRE1iMIR165a* line showed a relatively weaker suppression and much faster recovery of gene expression of all five HD-ZIP III members compared to that of the *35SiMIR165a* line (Fig 5A and 5B), it showed a similar impact on BCN female development and syncytium expansion (Fig 5D and 5E). These results may be due to different tissue-specific expression of *CRE1* and *35S* promoters in *Arabidopsis* roots, and/or the fact that the *CRE1* promoter is expressed at the infection site up to 5 dpi while the 35S promoter is down-regulated as early as 3 dpi [118,119]. Nevertheless, these results suggested that moderate suppression of HD-ZIP III gene expression in the stele, even within a short period of time at the beginning of BCN infection, is sufficient to suppress syncytium formation. HD-ZIP III gene knock-down also significantly reduced syncytium size despite incomplete suppression and quick recovery of HD-ZIP III gene expression (Fig 5A, 5B and 5E). Overall, these results suggest that HD-ZIP III family transcription factors are important for both initial syncytium establishment and syncytium expansion.

Recently, Smetana et al (2019) proposed that high auxin, and consequent expression of HD-ZIP III genes, promoted xylem identity and cellular quiescence in *Arabidopsis* root vasculature [64]. These xylem-identity cells serve as stem cell organizers, as they induce adjacent vascular cambial cells to divide and function as stem cells [64]. Knocking down or knocking out HD-ZIP III genes forces xylem parenchyma cells and cells with xylem identity to re-enter mitosis, causing excessive cell proliferation [64,67], while ectopic expression of ATHB8d in stem cells induced ectopic stem cell organizers and caused a vascular pattern shift [64]. Strong suppression of BCN infection in HD-ZIP III knock down lines suggests that HD-ZIP III

promotion of cellular quiescence is required for the transition of host cells to syncytial cells, either for initial syncytial cell establishment or syncytial cell incorporation. These quiescent cells with partial xylem identity probably serve as a transitional stage for the host cell to fully transform into a syncytial cell. In support of this hypothesis, Anjam et al (2020) observed that BCN appeared to always select cells next to the xylem for syncytium initiation [1], of which xylem adjacent (pro)cambial cells serves as stem cell organizer [64], and xylem pole pericycle cells are competent for lateral root initiation with high auxin levels [120]. Moreover, it has been reported that undifferentiated xylem precursor cells were always incorporated into the syncytium, buy not the fully differentiated xylem cells [2,121]. In our observation, we also noticed that when the syncytium is formed near the root tip, xylem differentiation was often disrupted (Figs 7A and S22A). All these observations point towards the idea that xylem-identity cells could be an important transitional stage for syncytium formation. After the cell enters this transitional stage, other nematode effectors, or factors from the host cell, will then be needed to inhibit further xylem differentiation, and divert the cell into a syncytium (S24 Fig). At this point, high auxin or HD-ZIP III expression would no longer be needed, and thus we saw down-regulation of *DR5::4xYFP* and *ProATHB8::4xYFP* expression in well-formed syncytium (Figs 8A–8C and S23A–S23C). Interestingly, ectopic protoxylem cells were often seen at the edge of developed syncytia (Fig 8A–8C), probably a result of insufficient signal in suppressing xylem differentiation at these locations.

The ectopic expression of *ATHB8d* in vascular cambium was sufficient to induce ectopic stem cell organizers, and to increase the number of xylem vessels [64]. Nevertheless, moderate overexpression of *ATHB8d-YFP* with either *ANT* or *35S* promoter (Fig 6B) was not sufficient to increase BCN infection rate or syncytial cell size (Figs 6D, 6E, S19B and S19C). It is possible that in such a scenario, factors that promote transition of the stem cell organizer to a syncytial cell become limiting elements for syncytium establishment or expansion. Strikingly, high expression of *ATHB8d-YFP* driven by *G1090* promoter also suppressed BCN infection (Figs 6B, 6D and S19B). Such an extremely high level of ATHB8d expression may have tipped the balance between stem cell organizer specification and its further differentiation, thus preventing it from transiting into a syncytial cell.

Overall, in this study, we identified *ATHB8* as a downstream factor of CLE signaling in BCN infection in *Arabidopsis*, and showed that *ATHB8*, along with other HD-ZIP III factors, play important roles in syncytium cell formation, probably by promoting host cells into a quiescent status (S24 Fig). However, the exact role of each individual member of HD-ZIP III in BCN infection, and their connection with CLE and auxin signaling would need further investigation.

## Materials and methods

### Plant materials

Plants are in Col-0 background unless otherwise specified. *clv1-101 clv2-101 rpk2-5* [58], *ProATHB8::GUS* (Ws) [122], *ProATHB8::4xYFP* [123], *Pro35S::XVE>>GUS*, *ProG1090:: XVE>>GUS* [124], *Pro35S::XVE>>MIR165a*, *ProANT::XVE>>ATHB8d-YFP*, *ProMIR165a:: GFP*, *ProMIR166b::GFP*, *MIR165/6-GFP* sensor line [67], and *ProCRE1::XVE>>MIR165a* [89], *athb8-11* (Col er-2 background), *cna-2* (Col er-2 background), *athb8-11 cna-2*, *phb-13* (Col er-2 background), *phv-11* (Col er-2 background), *phb-13 phv-11 cna-2*, and *rev-6* (Ler background) [87] have been described previously.

### Beet cyst nematode infection assay

*H. schachtii* was propagated on sugar beet (*Beta vulgaris* cv. Monohi). Nematode eggs were isolated and hatched as previously described [125]. After 2–3 days, J2s (2nd stage Juveniles) were

collected and surface-sterilized with 0.004% Mercuric Chloride, 0.004% Sodium Azide and 0.002% Triton X-100 solution for 7 min, washed with sterilized water for 6 times, and then resuspended in 0.05% agarose. *Arabidopsis* seeds were sterilized and grown on modified Knop's medium with Daishin agar (Brunschwig Chemie, Amsterdam, The Netherlands) [126].

For phenotypic analysis, seeds of each genotype were placed in 12 well plates following a random block design, stratified at 4 degree for 2–3 days, and were grown in a chamber set at 24°C with 12 h light/12 h dark light cycle. Fourteen days after germination, 180–200 J2s were inoculated on each root. The penetration rate was counted 3–5 days after inoculation. Number of J4s/mature females were counted on 14th day and 30th day after inoculation. For syncytia size measurement, syncytia associated with single female were imaged on 15th day after inoculation and syncytia sizes were measured with ImageJ. When Estradiol induction is needed, 10 mM Estradiol stock in DMSO was first diluted to 25x of the working concentration in sterile water pre-warmed to 60°C, and 100 ul of the diluted Estradiol or DMSO were added to each well (containing 2.5 ml Knop's medium). Estradiol induction starts 2 days before inoculation.

## Penetration assay

Penetration rate was counted at 3–5 days post inoculation. Shoots of seedlings were first removed by scissors. The solid Knop's media was melted by placing 12-well plates in a boiling water bath for 8–10 min and subsequently removed. Each well was rinsed with water twice to remove media residues. Nematodes were stained with boiling acid fuchsin solution for 1–2 min, rinsed with 95% ethanol once, and then kept in 95% ethanol till counting.

## Collection of infected root segments

To collect infected root segments, seeds were plated on square plates and grown vertically. About 10–15 sterilized *H. schachtii* J2s were inoculated on each root on the 7th day after germination, and root segments with syncytium (about 8 mm in length), or corresponding root segments on uninfected or HsCLE2p treated seedling roots, were cut under a stereoscope with scissors 4 days after inoculation. Root segments were immediately frozen in liquid nitrogen and were kept in -80°C freezer until RNA isolation.

## RNA isolation

RNA was isolated by NucleoSpin RNA Plant Kit (Macherey-Nagel 1806/001) according to manufacturer's instruction. RNA integrity was inspected by a bioanalyzer.

## Library construction and sequencing

RNA-seq library construction and sequencing were carried out in the University of Missouri DNA Core Facility. Libraries were constructed following the manufacturer's protocol with reagents supplied in Illumina's TruSeq mRNA stranded sample preparation kit. Briefly, sample concentration was determined by Qubit flourometer (Invitrogen) using the Qubit HS RNA assay kit, and the RNA integrity was checked using the Fragment Analyzer automated electrophoresis system. mRNA samples were enriched by poly-A enrichment and fragmented. Double-stranded cDNA was generated from fragmented RNA, and the index containing adapters were ligated to the ends. Amplified cDNA constructs were purified by addition of Axyprep Mag PCR Clean-up beads. Final construct of each purified library was evaluated using the Fragment Analyzer, quantified with the Qubit fluorometer using the Qubit HS dsDNA assay kit, and diluted according to Illumina's standard sequencing protocol for sequencing on the NextSeq 500.

### RNA seq data analysis

For all samples, adapter sequence was removed using cutadapt (v0.16) [127]. Reads were then mapped to *Arabidopsis thaliana* transcripts (Ensembl release 42) using Salmon (v0.12.0) [128]. Transcript counts were converted into gene counts using the Bioconductor package tximport (v1.10.1) [129]. Differential expression of genes was tested using the Bioconductor package DESeq2 (v1.22.2) [80]. Gene ontology enrichment was analyzed by Bioconductor package topGO (v2.34.0) [130] with *Arabidopsis* annotation package org.At.tair.db (v3.7.0) [131].

### qPCR

For qPCR validation, the same RNA samples for RNAseq were reverse transcribed using PrimeScript 1st strand cDNA Synthesis Kit (Cat# 6110A) according to the manual. qPCR was conducted with Applied Biosystem PowerUp SYBR Green master mix (cat# A25741) on a Bio-Rad CFX Connect Real-Time PCR System. 0.1 ul cDNA was used for each qPCR reaction. Because typical references genes including *ACT2*, *ACT8*, *GAPDH*, *GAPB*, *UBP22*, and *UBC21* showed differential gene expression in our RNA sequencing dataset, a new set of reference genes were tested and *ACO3* (AT2G05710) was selected as the reference gene in our experiment (S6B and S6C Fig). Primers used for qPCR were listed in S2 Table.

### GUS staining

For GUS staining of infection sites, seedlings were grown vertically on Knop's medium, and inoculated with about 10–15 sterilized *H. schachtii* J2s 5–7 days after germination. Infected roots were collected on the indicated date and stained with GUS staining solution (1mM K Ferricyanide, 1mM X-Gluc, pH 7.0) overnight at 37˚C.

### Confocal imaging

Infected roots were collected at indicated time points and fixed in 4% paraformaldehyde (PFA) at 4˚C overnight, washed twice with 1xPBS, and were then cleared with ClearSee solution [132] supplemented with 0.5 ug/ml Calcofluor White (Sigma, Cat# 18909-100ml) for at least one week before imaging.

For live imaging, reporter lines were grown on ½ MS medium for 4 days, then were transferred to Knop's media in a vertical plate. Each seedling was inoculated with about 10 sterilized *H. schachtii* J2s. About 18 hours after inoculation, seedlings that have been infected were selected under microscope and were moved to a block of Knop's media (with 100 ug/ml Timentin) on a microscope slide, and then were covered with a 22 x 40 coverslip. The coverslip was secured by tape on both ends to prevent it from moving. The assembly was put in a square plate with the coverslip facing down and kept in a growth chamber (24˚C, 12/12 light/dark cycle). GFP/YFP signal was imaged on a Zeiss LSM 880 confocal microscope 1–5 days after inoculation, at approximately the same time each day.

## Supporting information

**S1 Fig. Expression of the *WOX4* gene upon HsCLE2p treatment. A**. Response of *WOX4* gene expression to HsCLE2p treatment was quantified by qPCR to find an optimal time point for HsCLE2p treatment. The 180 min was selected as the treatment time for RNAseq samples. **B**. Induction of *WOX4* gene in RNA sequencing samples by qPCR.
(TIF)

**S2 Fig. Quality control of RNA-seq dataset. A.** Expression of *CLV1*, *CLV2*, *RPK2*, *WOX4*, and *CLE41* genes in the RNAseq dataset. Expression of *CLV1* and *CLV2* are barely detectable in the *clv* triple mutant due to *clv1-101* and *clv2-101* being null mutants. *RPK2* gene expression is comparable in the *clv* triple mutant and wild-type due to the *rpk2-5* allele being a point mutation (C3081A). **B.** PCA plot of RNAseq samples. All samples clustered as expected except for the "clv.CLE.1" sample, which was removed from subsequent differential gene expression analysis. **C.** Hierarchical clustering of RNAseq samples. Sample clv.CLE.1 (magenta) clustered with wild-type samples. The cluster is constructed using Euclidean distance with the 'complete' agglomeration method. **D.** The *clv* triple mutant showing the C3081A mutation (H1027Q on amino acid level) at the *RPK2* locus (AT3G02130) on the RNA-seq reads. Reads alignment was visualized using Integrated Genome Viewer (v2.11.3).
(TIF)

**S3 Fig. Overlapping genes down-regulated by BCN infection and HsCLE2p treatment in wild-type and the *clv* triple mutant. A.** Volcano plot of differentially expressed genes upon BCN infection and HsCLE2p treatment in wild-type and the *clv* triple mutant. **B—C.** Venn diagram showing overlap of up-regulated genes between wild-type and *clv* triple mutant upon BCN infection (B) or HsCLEp treatment (C). **D—E.** Venn diagram showing overlap of down-regulated genes between wild-type and the *clv* triple mutant upon BCN infection (**D**) or HsCLE2p treatment (**E**). **F, H.** BCN up-regulated (**F**) and down-regulated (**H**) genes in wild-type and *clv* triple mutant were enriched in similar GO terms of biological process. **G, I.** HsCLE2p up-regulated (**G**) and down-regulated (**I**) genes in wild-type and *clv* triple mutant were enriched in different GO terms for biological process.
(TIF)

**S4 Fig. BCN infection reduced gene expression differences between *clv* triple mutant and wild-type roots. A–B.** Venn diagram showing BCN infection reduced numbers of DEGs between the *clv* triple mutant and wild-type roots compare to control samples. **C–D.** HsCLE2p treatment increased numbers of DEGs between the *clv* triple mutant and wild-type roots compared to control samples. **E.** A diagram to visualize comparisons shown in panel A–D.
(TIF)

**S5 Fig. Filtering for downstream genes that are negatively regulated by CLE signaling.** A. Venn diagram showing number of genes that are 1), Up-regulated in the *clv* triple mutant compared to wild-type when uninfected. 2), Down-regulated by HsCLE2p treatment in the wild-type but not in the *clv* triple mutant (3). 4), Down-regulated by BCN infection in the wild-type. **B.** A diagram showing criteria used in (A). Red font color represents up-regulated sets; Blue font color represents down-regulated sets. Gray font color represents non-significant sets. n.s., not significant; *, $p < 0.05$.
(TIF)

**S6 Fig. Selection of a new set of qPCR reference genes. A.** Expression of published reference genes, *ACT2* [133], *ACT8* [134], *GAPDH* [135], *UBP22* [136, 137], *UBC21* [25], and *GAPB* [138], in the RNAseq dataset. **B.** Selection of a set of new candidate genes that are consistently expressed across all samples. **C.** qPCR primer efficiency of selected candidate genes. The *ACO3* gene is selected as the new reference gene for qPCR test.
(TIF)

**S7 Fig. qPCR verification of selected genes that are positively regulated by CLE signaling in BCN infection.** qPCR verification of a subset of tier 1 genes in S4 Dataset. Expression was verified for the rep1 and rep2 of RNAseq samples, with similar results. Only results from rep1

were shown here. Two technical replications were used for each sample. Letters above each bar graph represent statistical group of Tukey's HSD test following ANOVA analysis.
(TIF)

**S8 Fig. Expression of HD-ZIP III transcription factors in the RNA sequencing dataset. A.** Expression of HD-ZIP III transcription factors in the RNAseq dataset.
(TIF)

**S9 Fig. Expression of *ProATHB8::GUS* in 5-day old seedling.** *ProATHB8::GUS* expression in 5-day old uninfected wild-type (Ws-2). Areas of yellow boxes were shown in the right panel at higher magnification with corresponding codes. The expression of *ATHB8* gene along the root vasculature is not even.
(TIF)

**S10 Fig. Expression of *ProATHB8::4xYFP* in uninfected root and BCN infection site (5 dpi). A.** Expression of *ProATHB8::4xYFP* in 7-day-old uninfected root. A very weak YFP signal was seen in the mature vasculature (top panels) while strong YFP signal was detected in the root apical meristem (bottom panels). Insert, optical cross section of root tip showing *ProATHB8::4xYFP* was expressed in the xylem axis cells. **B.** Expression of *ProATHB8::4xYFP* at infection site at 5 dpi. Optical sections show concentration of YFP signal in the periphery of the syncytium. The root segment was fixed and cleared according to Kurihara *et al.*, 2015 [132]. The yellow dashed line indicates the position of the nematode. Red dashed lines indicate the location of optical cross section. Cyan, Calcofluor White. S, Syncytium. bar = 100 µm.
(TIF)

**S11 Fig. Quantification of *ProATHB8::4xYFP* signal in additional BCN infection sites. A-F**, Increasing of *ProATHB8::4xYFP* signal in six independent BCN infection sites. **G**. Quantification of YFP signal intensity at syncytia (s, red line) or adjacent sites (a, cyan line) in image **A-F** (labeled A'-F' respectively). Red arrowhead, position of nematode head. bar = 200 µm.
(TIF)

**S12 Fig. Expression of *ProMIR165a::GFP* and *ProMIR166b::GFP* in *Arabidopsis* root. A.** Expression of *ProMIR165a::GFP* in uninfected *Arabidopsis* root. GFP signal was detected in endodermis cells of both mature root (left panels) and the root tip (right panels). Optical cross section (red box) showed the expression of GFP in endodermis cells. **B.** Expression of *ProMIR166b::GFP* in endodermal cells at the root tip in uninfected root. GFP signal was not detected in mature root. **C**. *ProMIR166b::GFP* expression is not detected in BCN infection site. The same nematode was monitored from 1 to 5 days post inoculation (dpi). Green Signal near infection site is autofluorescence. **D.** the same seedling shown in (**C**) is GFP positive at the root tip (imaged at 5 dpi). Red dashed line, position of cross section. White arrowhead, position of nematode head. bar = 100 µm.
(TIF)

**S13 Fig. Quantification of *ProMIR165a::GFP* signal in additional BCN infection sites.** A-F, decreasing of *ProMIR165a::GFP* signal in six independent BCN infection sites. **G**. Quantification of GFP signal intensity at syncytia (s, red line) or adjacent sites (a, cyan line) in image A-F (labeled A'-F' respectively). Red arrowhead, position of nematode head. bar = 200 µm.
(TIF)

**S14 Fig. Expression of *MIR165/166* regulatory genes, *AGO1*, *AGO10*, *SHR*, and *SCR* in the RNAseq dataset.** AGO1, which recruits *MIR165/166* for target mRNA cleavage, is down-regulated. AGO10, which specifically sequesters *MIR165/166* for degradation, is up-regulated. SHR

is a stele expressed transcription regulator which moves to endodermis cells and interacts with SCR to activate *MIR165/6* expression.
(TIF)

**S15 Fig. Expression of *MIR165/6* sensor in BCN infection site in *Arabidopsis* root. A.** Expression of *MIR165/6* sensor in uninfected roots. Bar = 100 μm. B. Expression of *MIR165/6* sensor in the BCN infection site. The same infection site was monitored for five days. White arrowhead, position of nematode head. Bar = 100 μm for 1 and 2 dpi images, 200 μm for 3–5 dpi images.
(TIF)

**S16 Fig. Biological replicates of BCN infection assays on *athb8-11*, *cna-2*, and *athb8-11 cna-2* mutants.** Loss of *ATHB8* and/or its closest homologue *ATHB15/CNA* does not affect BCN penetration (A), female development (B), or syncytium size (C). Data from Rep 1 is shown in Fig 4D–4F. Bar graphs represent mean ± SE. Dots represent each individual measurement. Statistical tests were performed with Wald test following generalized linear mixed-effect model (for penetration and infection data) or linear mixed-effect model (for syncytium size).
(TIF)

**S17 Fig. Biological replicates of BCN infection assays on *Pro35SiMIR165a and ProCRE1i-MIR165a* inducible lines.** Inducible expression of *MIR165a* suppresses BCN female development (B) and syncytium size (**C**), but not BCN penetration (**A**) in *Arabidopsis*. Data from Rep 3 of (A-B) is shown in Fig 5C and 5D. Combined data from (**C**) is shown in Fig 5. Bar graphs represent mean ± SE. Dots represent each individual measurement. Statistical tests were performed with Wald test following generalized linear mixed-effect model (for penetration and infection data) or linear mixed-effect model (for syncytium size). Only statistically significant results were labeled.
(TIF)

**S18 Fig. Mutations of HD-ZIP III family member *PHB/ATHB14*, *PHV/ATHB9*, and *REV* do not affect BCN infection. A.** *phb-13*, *phv-11* single and *phb-13 phv-11 cna-2* triple mutants didn't show significant root growth defect in both vertical plates (left and middle panel) and 12-well plate (right panel). B. *rev-6* mutant did not show significant root growth defect in both vertical plates (left and middle panel) and 12-well plate (right panel). **C-E.** *phb-13*, *phv-11*, and *cna-2* mutants as well as their triple mutant didn't show significant different in BCN penetration (**C**), female development (**D**), and syncytium size (**E**) compared to that of wild-type (Col *er-2*). **F-H.** *rev-6* mutant didn't show significant different in BCN penetration (**F**), female development (**G**), and syncytium size (**H**) compared to that of wild-type (*Ler*) Statistical tests were performed with Wald test following generalized linear mixed-effect model (for penetration and infection data) or linear mixed-effect model (for syncytium size). bar = 1 cm in **A** and **B**.
(TIF)

**S19 Fig. Biological replicates of BCN infection assays on inducible *ATHB8d-YFP* overexpression lines.** Effect of inducible overexpression of *ATHB8d-YFP* on BCN penetration (**A**), female development (**B**), and syncytium size (**C**) in *Arabidopsis*. Data from Rep 1 is shown in Fig 6 (**C-E**). Bar graphs represent mean ± SE. Dots represent each individual measurement. Statistical tests were performed with Wald test following generalized linear mixed-effect model (for penetration and infection data) or linear mixed-effect model (for syncytium size). Only

statistically significant results were labeled.
(TIF)

**S20 Fig. Infection of *ATHB8d-YFP* overexpression lines in the vertical plates.** Seedlings were grown on vertical plates with Knop's media for 5 days, then were moved to Knop's with DMSO or 5 µM Estradiol. Two days after estradiol induction, each root was inoculated with about 15 J2s of BCN. Plates were imaged five days after inoculation. The *ProG1090iATHB8d-YFP* line showed distorted root development after estradiol induction with or without BCN infection. bar = 1cm.
(TIF)

**S21 Fig. Above-ground phenotype of *ATHB8d-YFP* overexpression plants in 12-well plates.** 18 seedlings for each genotype/treatment combination were grown in 12-well plates with Knop's media. 5 µM estradiol or an equivalent amount of DMSO were added to each well at 12 days post germination. Two days after estradiol induction, each well was inoculated with about 200 J2s of BCN. Plates were imaged 5 days after inoculation. Images of 10 seedlings from each genotype/treatment combination were shown. E2, estradiol.
(TIF)

**S22 Fig. Expression of auxin reporter *DR5::4xYFP* in the BCN induced syncytium at 1 dpi.** A. A confocal optical section of developing syncytium with *DR5::4xYFP* expression. B. Optical cross sections of positions shown in (A). White dashed line, outline of nematode head. White arrow, position of the stylet. Red dashed line, positions of cross section shown in (B). Yellow dashed line, outline of the syncytium. Red arrowhead, disrupted xylem differentiation. x, xylem cells. White dot, pericycle cells. White circle, pericycle cells incorporating into the syncytium. CFW, calcofluor white staining of cell walls. YFP, yellow fluorescent signal. bar = 20 µm.
(TIF)

**S23 Fig. Distribution of *DR5::4xYFP* activity at the BCN infection site at 3 dpi.** White dashed line, outline of nematode. Red box, zoomed in portion shown in (B). Red dashed lines in (B), position of optical cross sections shown in (C). Yellow dashed line, outline of the syncytium. x, xylem cells. CFW, calcofluor white staining of cell walls. YFP, yellow fluorescent signal. bar = 50 µm.
(TIF)

**S24 Fig. A conceptual model showing *ATHB8* as a downstream factor of the *Arabidopsis* CLE signaling pathway functioning with other HD-ZIP III genes in promoting both initial syncytium formation and syncytium expansion.** High HD-ZIP III gene expression promotes plant cells into quiescent status with partial xylem identity known as a stem cell organizer, indicating that the nematode may use a stem cell organizer as an intermedia status for syncytium initiation/incorporation. Other BCN effectors may be needed to suppress xylem differentiation and/or promote the transit of a stem cell organizer into a syncytial cell.
(TIF)

**S1 Table. Mapping efficiency of RNA sequencing reads.**
(XLSX)

**S2 Table. Primers used for qPCR.**
(XLSX)

**S1 Dataset. RNA-seq count table.**
(TXT)

**S2 Dataset. RNA sequencing differential gene expression analysis—All comparisons.**
(XLSX)

**S3 Dataset. Gene Ontology (Biological Process) analysis of differential expressed genes.**
(XLSX)

**S4 Dataset. List of genes positively regulated by CLE signaling in BCN infection.**
(XLSX)

**S5 Dataset. List of genes negatively regulated by CLE signaling in BCN infection.**
(XLSX)

## Acknowledgments

Special thanks to Clinton Meinhardt, Amanda Howland, Ben Averitt, Dean Kemp, and Kurk Lance for nematode population maintenance throughout the span of this project. We thank Dr. Nathan Bivens from University of Missouri DNA Core for his assistance in RNA seq library preparation. We thank Dr. Bill Spollen from University of Missouri Informatics Research Core for his assistance with RNA seq data analysis. We thank Dr. Muthugapatti K. Kandasamy of UGA Biomedical Microscopy Core for his confocal microscopy expertise and assistance. We thank Drs. Ari Pekka Mähönen and Annelie Carlsbecker for providing seeds.

## Author Contributions

**Conceptualization:** Xunliang Liu, Melissa G. Mitchum.

**Data curation:** Xunliang Liu.

**Formal analysis:** Xunliang Liu.

**Funding acquisition:** Melissa G. Mitchum.

**Investigation:** Xunliang Liu, Melissa G. Mitchum.

**Methodology:** Xunliang Liu.

**Project administration:** Melissa G. Mitchum.

**Resources:** Melissa G. Mitchum.

**Supervision:** Melissa G. Mitchum.

**Writing – original draft:** Xunliang Liu.

**Writing – review & editing:** Melissa G. Mitchum.

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
