## [Decision Letter · Decision Letter 0]

25 May 2024

Dear Dr. Mitchum,

Thank you very much for submitting your manuscript "A Major Role of Class III HD-ZIPs in Promoting Sugar Beet Cyst Nematode Parasitism of Arabidopsis" for consideration at PLOS Pathogens. As with all papers reviewed by the journal, your manuscript was reviewed by members of the editorial board and by several independent reviewers. In light of the reviews (below this email), we would like to invite the resubmission of a significantly-revised version that takes into account the reviewers' comments.

We cannot make any decision about publication until we have seen the revised manuscript and your response to the reviewers' comments. Your revised manuscript is also likely to be sent to reviewers for further evaluation.

Sincerely,

Savithramma P. Dinesh-Kumar

Section Editor

PLOS Pathogens

Savithramma Dinesh-Kumar

Section Editor

PLOS Pathogens

Michael Malim

Editor-in-Chief

PLOS Pathogens

orcid.org/0000-0002-7699-2064

Reviewer's Responses to Questions

**Part I - Summary**

Reviewer #1: The manuscript by Liu et al. identifies and characterizes ATHB8, which encodes a HD-ZIP III family transcription factor, as a downstream component of the CLE signaling pathway in syncytium formation. ATHB8 is expressed in the syncytium during the early stages of infection and then transitions to neighboring cells of the syncytium as it expands, showing an expression pattern coincident with auxin response at the infection site. Knocking down HD-ZIP III by over-expressing MIR165a reduced the female development of the cyst nematode. Overall, the manuscript is well-written, the data seems rigorous, and it provides an important advance in understanding cyst nematode-induced syncytium formation.

Reviewer #2: In this manuscript, Liu et al. analyzed the role of the CLE signaling pathway in syncytium formation and revealed the function of HD-ZIP III TFs, including ATHB8, in connecting the CLE and auxin signaling pathways to promote syncytium formation. The authors provide an interesting and well written paper. Their dataset is informative and communicated in the context. However, some results need to be improved. My comments on the manuscript are as follows.

Reviewer #3: Manuscript entitled ‘A Major Role of Class III HD-ZIPs in Promoting Sugar Beet Cyst Nematode Parasitism of Arabidopsis’ identified ATHB8, encoding a HD-ZIP III family transcription factor, as a downstream component of the CLE signaling pathway in syncytium formation, and knockdown of HD-ZIP III TFs by inducible over-expression of MIR165a in Arabidopsis dramatically reduced female development of the sugar beet cyst nematode. The authors obtained some promising experimental data.

Reviewer #4: In this work, the authors investigated the potential function of Class III HD-ZIP TF genes in promoting sugar beet cyst nematode (SBCN) infection of Arabidopsis roots. Based on the evidence obtained, they suggest that Class III HD-ZIP TF genes play a major role for efficient parasitism of Arabidopsis by SBCN. The major strength is that the authors endeavored to overcome functional redundancy of HD-ZIP TF genes during their regulation of SBCN infection. The weakness of the work is that silencing the expression of Class III HD-ZIP TF genes by inducible expression of MIR165a had highly variable effects on SBCN infection of Arabidopsis, which do not support the claim that Class III HD-ZIP TF genes play “a major role” in promoting SBCN parasitism of Arabidopsis.

**Part II – Major Issues: Key Experiments Required for Acceptance**

Reviewer #1: (1) The authors used a time point of 4 days after inoculation for their transcriptome study. It would be helpful if the authors could explain why they chose this specific time point.

(2) Lines 207-214: The authors suggest that the transcriptomic similarity between the clv triple mutant and wild-type samples in response to BCN infection indicates that BCN-infected samples might not be optimal for filtering out downstream genes of the CLE signaling pathway. Instead, they propose that HsCLE2p treatment would be a more effective approach. While I understand the rationale, I would argue that a reduced number of DEGs between the clv triple mutant and wild-type is advantageous as it highlights fewer but biologically relevant differences. In contrast, exogenous peptide treatment, where all root tissues are treated with a peptide at a potentially non-physiological concentration, might not provide the same relevance. Perhaps the authors could clarify this point further.

(3) Line 221: Including this methodology in Figure 2 would be helpful to readers.

(4) Lines 300-309: Upon BCN infection, the expression of ProMIR165a::GFP signal is gradually reduced at the infection site. I observed expression only in endodermal cells around the feeding site, which seems to disappear after 5 days. The authors should clarify the significance of this expression pattern more thoroughly. In general, I found the expression pattern of MIR165a::GFP a bit puzzling.

(5) Figure 4a vs. 4b: There is a difference in the expression pattern between Figures 4a and 4b. The authors should address this difference explicitly in results section i.e. MIR165a expression in endodermis and transitioning..

(6) Figure 5: Please add to the figure legend how many times the experiment was independently replicated.

Reviewer #2: 1. What is the best time point at which BCN induces the most significant DEGs? The authors need to test this by experiments, or it is not clear to the readers whether the less DEGs results from no or over induction by BCN in Figure 1. The authors need to discuss this at least.

2. The authors analyzed the protein localization using transgenic plants, however, whether these fused protein are expressed correctly remains unclear. They should test both the mRNA expression level and the protein level in the transgenic plants to confirm their results that the localization or accumulation are affected by the BCN. For example, the correct expression of ATHB8-YFP is very important for drawing conclusions.

3. Line 363: These results are confusing. The expression of HD-ZIP III TFs was not significantly downregulated, but the BCN infection phenotype was very obvious. How to explain the phenotype vs. gene expression levels? If so, the athb8 cna double mutant should show the same phenotype with no difference in infection rate.

Reviewer #3: However, the results show that loss of function of ATHB8 and ATHB15/CAN does not affect BCN infection, and overexpression of ATHB8 does not increase susceptibility of Arabidopsis to BCN, meaning ATHB8 might not be associated with BCN susceptibility. If so, although auxin level and ATHB8 expression are elevated in neighboring cells of the developing syncytium, it is difficulty to conclude that HD-ZIP III TFs might function as a connecting point for CLE and auxin signaling pathways in promoting syncytium formation, possibly by inducing root cells into a quiescent status and priming them for initial syncytial cell establishment and/or subsequent cellular incorporation. I suggest authors continue to test the other HD-ZIP III TFs members which contribute(s) to BCN susceptibility in Arabidopsis. There are a lot of work to be done.

Reviewer #4: 1) The authors used the number of SBCN females per plant at the lates stage of infection as a parameter to assess the effects of silencing HD-ZIP TF genes, and they conclude that silencing HD-ZIP TF genes impairs syncytium formation. However, they should examine SBCN infection sites per plant at the early and middle stages of the infection, e.g., 1, 3, 4, 6, 12 days after inoculation, to reveal if lowering HD-ZIP TF gene may indeed impair the initiation and/or early development of syncytium.

2) The authors suggest that upregulation of HD-ZIP TF genes by SBCN infection may facilitate early syncytium formation/development possibly by inducing root cells into a quiescent status. To support this proposition, the authors are advised to use suitable biomarkers to check if cellular quiescence may indeed occur in the roots at the early or middle stages of SBCN infection.

**Part III – Minor Issues: Editorial and Data Presentation Modifications**

Reviewer #1: Including a model at the end of the manuscript would help explain the findings to the audience.

Reviewer #2: 1. Line 237 and Figure S6: Why ATHB8 is a strong candidate that was chosen by the authors? The change in ATHB8 expression was not significant compared to that of several other genes.

2. Figure 4: The control should be added in this Figure. Additionally, for clarification, quantitative analysis of YFP/GFP signals should be performed.

3. Figure 6B and 6D: The expression of ATHB8 and its homologs was not suppressed at 14 dpi, but the number of females in these plants also decreased. Why?

4. All Figures should be referenced in order of appearance.

5. Line 187: The number was wrong. 30.98% (790 out of 2,550)?

6. Line 192: The recognition of CLE-like effectors by CLVs should leads to more DEGs, so why there is no significant difference between the wild type and mutants? Is CLV1/2 not important or not the main receptors?

7. Line 214: Moreover, BCN infection may suppress the expression of downstream genes by secreting some effectors?

8. Figure 2C and 2D: The same figures are presented in Figures S7 and S8. This is not standard. The same image should appear only once in the text.

9. Lines 274-290: Moving this paragraph to the Discussion section may be better. Moreover, the logic and readability of this paragraph is not strong.

10. Line 296: No difference of MIR165/166 expression has been reported previously, so why the authors chose MIR165/166? Removing this statement may be better.

11. Lines 358-359: The numbers come from different experimental replicates are not suggested to do quantitative analysis.

12. Line 440: Under the premise of functional redundancy, are other members doing the same to ATHB8? The authors may need to give a conclusion sentence that matches the Title in the manuscript.

13. The discussion section is too descriptive. The authors should discuss the most significant results at different points. The relationship between ATHB8 and the CLE signaling pathway should be further discussed.

14. The citation format of References needs to be unified.

Reviewer #3: Lines 334-344, explained why knockdown of HD-ZIP III TFs by inducible over-expression of MIR165a in Arabidopsis, but did not check expression level of the five HD-ZIP III TFs.

There are too much RNA-Seq data analyses in this manuscript. Suggest to move Fig. 1A-D to supplement data, and Fig. 1 and Fig. 2 merge into one figure.

Fig. 3 shows the expression of ATHB8 was decreased in WT, and the mutants used GUS, the result may not be true; in addition with L289-290 ‘the GUS staining technique may not be sensitive enough to distinguish such a small difference in gene expression (Fig. 3A-J)’, I suggest authors add the GUS stain area for analysis, or conduct ATHB8 expression analysis at each stage after BCN infection by qPCR.

Does Fig.4 show the same root? The size are different, the bar needs to be added in each image.

Fig. 6, The caption is wrong, it is for nematode development, but not for infection.

Fig. 8, The image is not clear, I suggest authors add ‘Bright Field’ single channel.

Supplemental figures, all ‘um’ are replaced with ‘μm’.

Reviewer #4: Because syncytium size was not significantly decreased by silencing HD-ZIP TF genes, it is difficult to understand the authors’ suggestion that HD-ZIP TF genes may be involved in cellular incorporation during syncytium formation in the roots with HD-ZIP TF gene expression lowered.

PLOS authors have the option to publish the peer review history of their article (what does this mean?). If published, this will include your full peer review and any attached files.

Reviewer #1: No

Reviewer #2: No

Reviewer #3: No

Reviewer #4: No
---

## [Decision Letter · Decision Letter 1]

21 Sep 2024

Dear Dr. Mitchum,

We are pleased to inform you that your manuscript 'A Major Role of Class III HD-ZIPs in Promoting Sugar Beet Cyst Nematode Parasitism in Arabidopsis' has been provisionally accepted for publication in PLOS Pathogens.

Best regards,

Savithramma P. Dinesh-Kumar

Section Editor

PLOS Pathogens

Savithramma Dinesh-Kumar

Section Editor

PLOS Pathogens

Michael Malim

Editor-in-Chief

PLOS Pathogens

orcid.org/0000-0002-7699-2064

Reviewer Comments (if any, and for reference):

Reviewer's Responses to Questions

**Part I - Summary**

Reviewer #1: (No Response)

Reviewer #2: All my concerns have been addressed by the authors.

**Part II – Major Issues: Key Experiments Required for Acceptance**

Reviewer #1: (No Response)

Reviewer #2: (No Response)

**Part III – Minor Issues: Editorial and Data Presentation Modifications**

Reviewer #1: (No Response)

Reviewer #2: (No Response)

PLOS authors have the option to publish the peer review history of their article (what does this mean?). If published, this will include your full peer review and any attached files.

Reviewer #1: No

Reviewer #2: No

---

## [Editor Report · Acceptance letter]

7 Oct 2024

Dear Dr. Mitchum,

We are delighted to inform you that your manuscript, "A Major Role of Class III HD-ZIPs in Promoting Sugar Beet Cyst Nematode Parasitism in Arabidopsis," has been formally accepted for publication in PLOS Pathogens.

Best regards,

Michael Malim

Editor-in-Chief

PLOS Pathogens

orcid.org/0000-0002-7699-2064